# Tail proteins of phage SU10 reorganize into the nozzle for genome delivery

Marta Šiborová [1], Tibor Füzik [1], Michaela Procházková [1], Jiří Nováček[1], Martin Benešík[2], Anders S. Nilsson [3] & Pavel Plevka [1] ✉

*Escherichia coli* phage SU10 belongs to the genus *Kuravirus* from the class *Caudoviricetes* of phages with short non-contractile tails. In contrast to other short-tailed phages, the tails of Kuraviruses elongate upon cell attachment. Here we show that the virion of SU10 has a prolate head, containing genome and ejection proteins, and a tail, which is formed of portal, adaptor, nozzle, and tail needle proteins and decorated with long and short fibers. The binding of the long tail fibers to the receptors in the outer bacterial membrane induces the straightening of nozzle proteins and rotation of short tail fibers. After the re-arrangement, the nozzle proteins and short tail fibers alternate to form a nozzle that extends the tail by 28 nm. Subsequently, the tail needle detaches from the nozzle proteins and five types of ejection proteins are released from the SU10 head. The nozzle with the putative extension formed by the ejection proteins enables the delivery of the SU10 genome into the bacterial cytoplasm. It is likely that this mechanism of genome delivery, involving the formation of the tail nozzle, is employed by all Kuraviruses.

Phages from the genus *Kuravirus* belong to the class *Caudoviricetes* of phages with short non-contractile tails[1]. Kuraviruses, including the *Escherichia coli* phage SU10, are distinguished among short-tailed phages by large genomes with 75–80,000 base pairs encoding more than 50 proteins [2–4]. Kuraviruses are lytic, and some of them have short replication cycles and produce a plentiful progeny, which makes them candidates for use in phage therapy against pathogenic strains of *E. coli*[5–7]. A phage cocktail containing phage ES17, from the genus *Kuravirus*, was used in a case study to treat *E. coli* infection of the prostate and urinary tract[8].

The genomes of kuraviruses are accommodated in prolate heads, characterized by fivefold symmetry and elongation in the direction of the tail axis[2,3]. The genomes of tailed phages are packaged into pre-formed pro-heads by molecular motors through a channel formed by the dodecamer of portal proteins, which replaces a pentamer of capsid proteins at one of the fivefold vertices of the capsid. Phage tails are attached to the portal complexes[9]. Common tail components of phages with short tails infecting Gram-negative bacteria are adaptor proteins, major tail proteins, tail needle proteins, and tail fibers[10]. The tail fibers enable the initial binding of phages to bacteria[11–13], whereas tail needles are responsible for the penetration of the host cell's outer membrane[14,15]. The heads of most short-tailed phages contain ejection proteins, which enable the delivery of phage genomes into the host cells. The ejection proteins form a translocation complex that elongates the tail to span the bacterial cell wall [16–19]. The pressure inside the phage head enables the ejection of 30–50% of the genome into a bacterial cell[20–22]. However, after equalization of the pressures inside the phage head and bacterial cytoplasm, the remainder of the DNA has to be delivered into the bacterium by another mechanism[21,23,24]. It has been shown, using negative-stain electron microscopy, that phage SU10 has a longer tail than most phages formerly classified to the family *Podoviridae*, and that the tail undergoes a conformational change upon binding to bacteria[2].

Here we present cryo-electron microscopy (cryo-EM) structures of the virion and genome release intermediate of bacteriophage SU10 and cryo-electron tomography (cryo-ET) characterization of its

[1]Central European Institute of Technology, Kamenice 753/5, 625 00 Brno, Czech Republic. [2]Faculty of Science, Masaryk University, Kamenice 753/5, 625 00 Brno, Czech Republic. [3]Department of Molecular Biosciences, The Wenner-Gren Institute, Stockholm University, 106 91 Stockholm, Sweden. ✉e-mail: pavel.plevka@ceitec.muni.cz

attachment to *E. coli* cells. After binding to the host, SU10 tail, nozzle proteins, and short fibers re-arrange to form a nozzle that extends the tail. Kuraviruses share more than 70% sequence similarity in their tail proteins. Therefore, it is likely that this mechanism of genome delivery, involving the formation of the tail nozzle, is employed by most, if not all Kuraviruses.

## Results and discussion
### Structure of SU10 virion
The virion of bacteriophage SU10 has a prolate head with a length of 1430 Å and a diameter of 460 Å, which is formed by the major capsid protein (gp9) (Fig. 1a, Supplementary Table 1, 2)[2]. The tail is 220 Å long and decorated with six long and six short tail fibers (Fig. 1a–c, 2a, c, Supplementary Table 1, 2). The tail is attached to one of the vertices of the prolate capsid, where the dodecamer of portal proteins (gp6) replaces a pentamer of capsid proteins (Figs. 1a–c). The part of the portal complex protruding from the capsid enables the binding of the dodecamer of adaptor proteins (gp11). The adaptor complex provides six attachment sites for long tail fibers, each of which is formed by a trimer of long tail fiber proximal proteins (gp12) and a trimer of long tail fiber distal proteins (gp13) (Figs. 1c, 2a, c). A hexamer of nozzle proteins (gp17) is attached to the adaptor complex and enables the binding of six short tail fibers, trimers of short tail fiber protein (gp16) (Fig. 2a, c). The tail needle, which is formed by three polypeptide chains (gp18), protrudes from the central channel formed by the nozzle proteins (Figs. 1a–c and 2a, c).

### SU10 capsid and structure of major capsid protein
The prolate capsid of SU10 is organized with fivefold symmetry and is built from 715 copies of the major capsid protein organized with the triangulation parameters $T_{end} = 4$, $T_{mid} = 20$ (Fig. 3a, Supplementary Fig. 1, 2). The major capsid protein forms capsomers of two types: pentamers and hexamers. The central tubular part of the capsid is built from 90 hexamers arranged in nine layers of a ten-entry helix (Fig. 3a). The capsid is closed on both ends with caps formed by the major capsid proteins organized with local T = 4 quasi-icosahedral symmetry. The closure of the caps is enabled by pentamers, which are bent in contrast to the flatter hexamers (Fig. 3a, Supplementary Fig. 1). However, the formation of the prolate SU10 head also requires variation in the shape of hexamers and in the angles at which they interact. The hexamers in the caps are more bent (H4, H5; 149-150°), whereas those that form the tubular part of the capsid are relatively flat (H1, H2L, H2R, H3; 154–164°) (Supplementary Fig. 2). All hexamer-hexamer and hexamer-pentamer interfaces in the caps are arched (145°) (Supplementary Fig. 2A), however, the hexamers that form the tubular part of the capsid are connected by two types of interfaces, arched (145°) and more planar (160°) (Supplementary Fig. 2). The arched interfaces enable the formation of rings of hexamers, whereas the more planar ones mediate interactions between the rings (Supplementary Fig. 2).

The major capsid protein of SU10 has an HK97 fold, which is common among tailed bacteriophages and herpesviruses (Fig. 3b)[25]. According to the convention, the capsid protein can be divided into the N-terminal arm, peripheral domain, extended loop, axial domain, and glycine-rich loop (Fig. 3b). The major difference between the

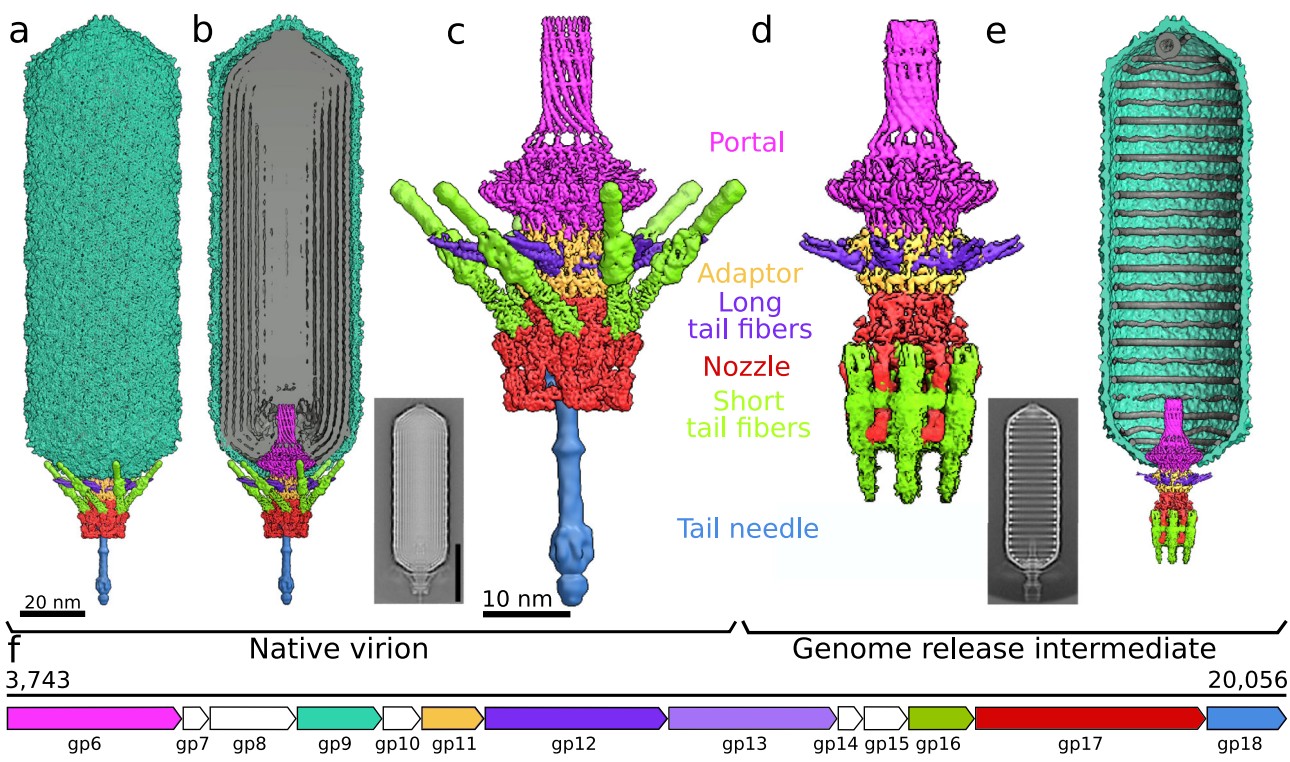

**Fig. 1 | Structure of virion and genome-release intermediate of phage SU10.**
**a** Surface representation of composite cryo-EM map of virion of phage SU10 colored according to protein type. The major capsid protein (gp9) is shown in turquoise, portal protein (gp6) in magenta, adaptor protein (gp11) in yellow, long tail fibers (gp12) in violet, nozzle protein (gp17) in red, short tail fibers (gp16) in green, and tail needle (gp18) in light blue. The length of the virion is 1590 Å. For details on the construction of the composite map, please see the Materials and methods section. **b** The same as A, but the front half of the composite map of the SU10 head was removed to show the structure of the genome in grey. The inset shows a 2D class average of the SU10 virion. The scale bar indicates 45 nm.

**c** Composite cryo-EM map of portal and tail complexes of SU10 virion. The length of the complex is 540 Å. **D** Cryo-EM reconstruction of portal and tail complexes from an SU10 genome release intermediate. **e** Composite cryo-EM map of genome-release intermediate of SU10. The front half of the head was removed to show the structure of the genome remaining in the capsid. The inset shows a 2D class average of the SU10 genome release intermediate. **f** Schematic representation of segment of SU10 genome encoding structural proteins color-coded the same as the proteins in panels **a** to **e**. Proteins shown in white are either non-structural or were not identified in the reconstructions.

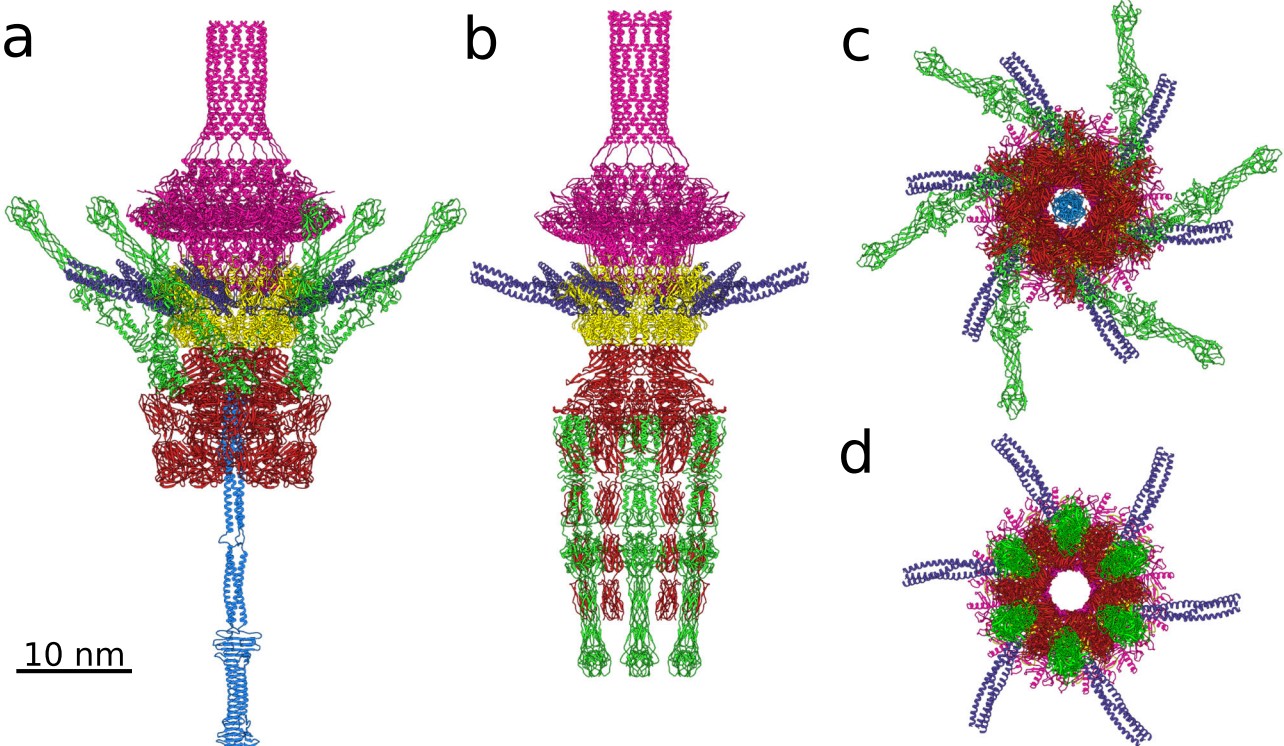

**Fig. 2 | Conformational changes of SU10 tail associated with genome release.** Cartoon representations of proteins forming tail of SU10 virion **a**, **c** and genome release intermediate **b**, **d**. The portal protein (gp6) is shown in magenta, adaptor protein (gp11) in yellow, long tail fiber protein (gp12) in violet, nozzle protein (gp17) in red, short tail fiber protein (gp16) in green, and tail needle protein (gp18) in light blue. **a**, **b** views perpendicular to the tail axis and **c**, **d** views along the tail axis towards the phage head. In the SU10 virion, the short tail fibers are tilted towards the capsid and the tail needle protrudes from the tail complex. The genome release intermediate of SU10 **b**, **d** lacks the tail needle and contains a 277 Å-long nozzle formed by short tail fibers and nozzle proteins. Tail needle and central and receptor-binding domain of the short tail fiber were modeled using AlphaFold2 multimer and fitted into the cryo-EM map. The remaining structures were built into cryo-EM reconstructions. For details on the structure prediction and building, please see the Materials and methods section.

capsid proteins forming hexamers and pentamers is in the structure of the annular loop positioned at the tip of the axial domain (Fig. 3b). In hexamers, the annular loop protrudes away from the axial domain (Fig. 3b, c). The six annular loops from subunits forming one hexamer form a ring around its quasi-sixfold axis (Fig. 3c). In contrast, in subunits forming pentamers the annular loop is bent 90° towards the capsid center (Fig. 3c). The beta-strands from the loops of major capsid proteins from a pentamer form a beta-barrel with a central pore with a diameter of 8 Å that penetrates through the capsid. (Fig. 3c, Supplementary Fig. 3G). The pore may serve for the passage of ions and water molecules from and into the capsid during genome packaging and ejection.

The genome of SU10 encodes a protein (gp10) with a predicted immunoglobulin-like fold similar to that of the minor capsid protein of phage Epsilon15 and the N-terminal domain of head fiber protein of bacteriophage φ29[26,27]. However, the cryo-EM reconstruction of the SU10 head does not contain density corresponding to minor capsid proteins. It is possible that gp10 of SU10 has a different function or binds to the capsid with low affinity and was lost during phage purification (Supplementary Fig. 4C).

### Structure of genome packaged in SU10 virion

The cryo-EM density of the genome in the SU10 virion was resolved in both fivefold symmetric and asymmetric reconstructions of its head (Fig. 1b, Supplementary Fig. 3). Despite the differences in the symmetries imposed during the reconstruction processes, the arrangements of the genome densities are similar (Supplementary Fig. 3A–D). Nine shells of density can be distinguished inside the tubular part of the head, whereas only four shells are resolved underneath the capsid caps (Supplementary Fig. 3). In the three outermost shells, the density separates into strands, which can be interpreted as segments of double-stranded DNA (Supplementary Fig. 3A–D, F). The resolved structure of the genome in the cryo-EM reconstruction indicates that the genome packaging results in a similar ordering of the DNA in most SU10 virions. However, links between the DNA segments in the central part of the SU10 head, which must exist because the genome is formed of a single 77,325 bp long double-stranded DNA[2], are not resolved. In both five-fold symmetric and asymmetric reconstructions of the SU10 head, the DNA strands in the two outermost layers are organized as ten-entry helices (Supplementary Fig. 3B, D). The DNA strands from the outermost layer of the genome curve to avoid the beta-barrels protruding from pentamers of capsid proteins (Supplementary Fig. 3G). The intriguing organization of the DNA may be used as additional restraints in future studies of the phage genome packaging process by computer simulations.

The proteins gp20-24 are present in the SU10 virion, but absent from the genome release intermediate, indicating that they are ejection proteins (Supplementary Fig. 4C, Supplementary Table 1). However, the reconstruction of the SU10 virion does not contain resolved density that could be attributed to the putative ejection proteins (Fig. 1b), as is the case in tailed phages T7, Epsilon15, and N4[17,18,28]. The genome of SU10 is packaged in the capsid with a density of 0.48 Da/Å$^3$, which is comparable to the 0.49–0.52 Da/Å$^3$ determined previously for bacteriophages P22, lambda, T4, φ29, and T7[29]. Therefore, the capsid of SU10 could contain both genomic DNA and ejection proteins.

### Portal complex

The portal complex of SU10 is embedded in the capsid at one of its fivefold vertices, where it replaces a pentamer of capsid proteins (Figs. 1a, b, 4a). The portal protein of SU10 can be divided into the wing,

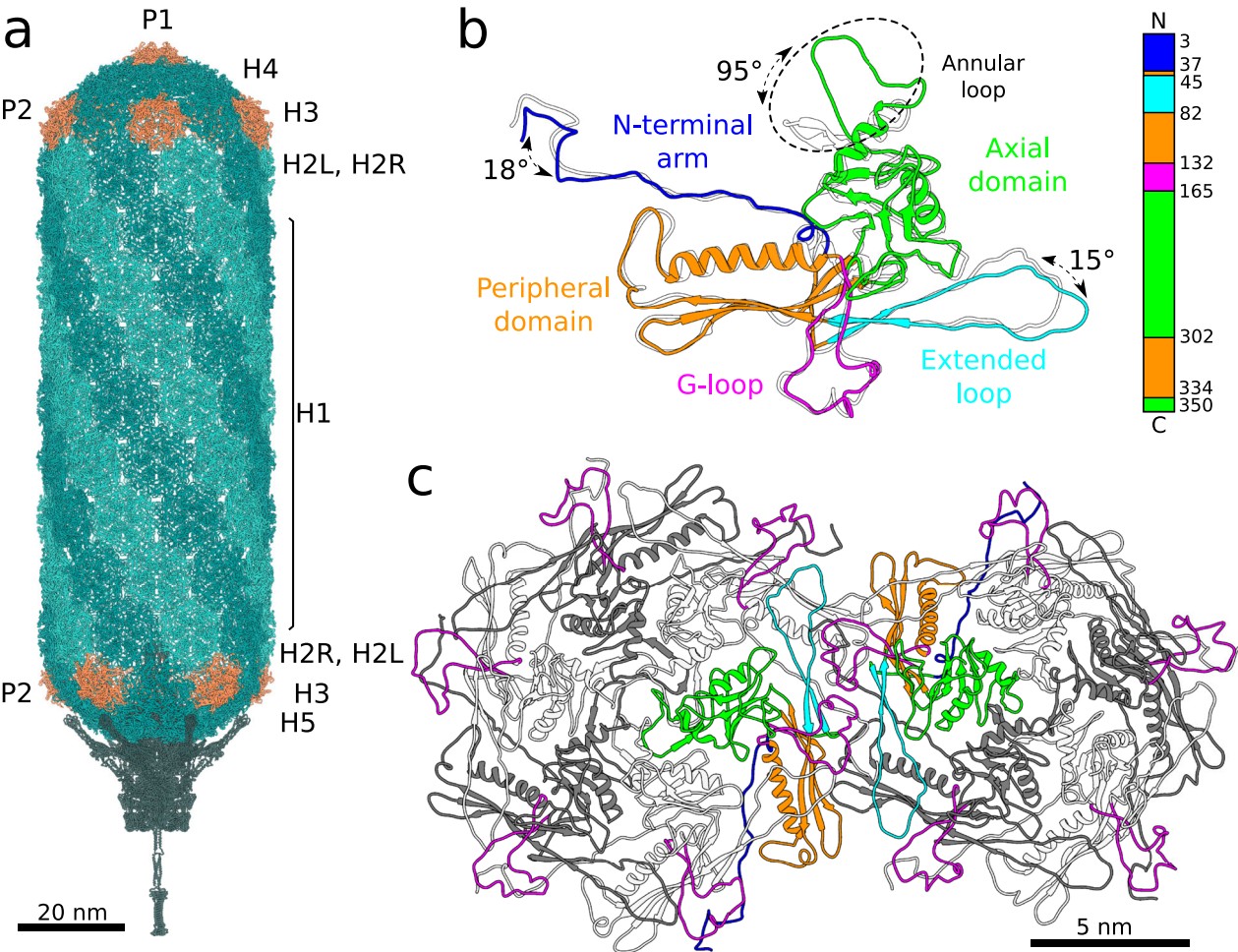

**Fig. 3 | SU10 capsid and major capsid protein. a** Cartoon representation of proteins forming virion of phage SU10. The prolate capsid has fivefold symmetry and the capsid proteins are organized with the parameters of $T_{end} = 4$, $T_{mid} = 20$. Pentamers of capsid proteins are shown in orange, hexamers in dark cyan or light sea green. Pentamers are of two types, based on their position in the capsid. The P1 pentamer is positioned on the fivefold axis of the particle, whereas the remaining P2 pentamers are positioned on quasi-fivefold axes at the borders between the tubular part of the capsid and the caps. There are six variants of hexamers: H1 that form the central part of the prolate head; H2L and H2R belong to the central part of the capsid but interact with terminal caps; H3 are between P2 pentamers; H4 are between P1 and P2 pentamers; and H5 are between P2 pentamers and the portal

complex. **b** Comparison of structures of major capsid proteins forming H1 hexamer (colored according to the domains) and P1 pentamer (white). The major capsid protein of SU10 has an HK97 fold[25,66]. The major differences in the structures of P1 and H1 subunits are in the N-terminal arm, extended-loop, and the annular loop. The domain organization of the major capsid protein is shown as a 1D plot. **c** Cartoon representation of major capsid proteins forming P2 pentamer and H2R hexamer. Two major capsid proteins that mediate the majority of contacts between the pentamer and hexamer are shown with domains highlighted in color. The remaining subunits are shown in alternating white and dark grey. The G-loops of all subunits are highlighted in magenta.

stem, clip, crown, and barrel domains (Fig. 4b)[29]. The wing domain is the largest, and is formed by eight helices and a β-sandwich containing two β-sheets of five and three antiparallel β-strands (Fig. 4b). The positively charged side-chains of Lys[3] and Lys[5] from the wing domain interact with the DNA that wraps around the portal complex (Fig. 4a). The two lysines are repeated twelve times in the portal complex, and thus form a high-avidity interface that is likely to bind DNA soon after the genome packaging is initiated. Anchoring the DNA end to the portal would influence the packaging of the genome into the head and its final structure. Furthermore, the wing domain mediates the interactions of the portal complex with the capsid (Fig. 4a, c). Because of the mismatch of the fivefold symmetry of the capsid and twelvefold symmetry of the portal, we characterized their interactions using an asymmetric reconstruction of the neck region of the SU10 virion, which reached a resolution of 4.6 Å (Supplementary Table 2). Residues 8–44 from the wing domain form a curved α-helix that lines the inner face of the capsid (Fig. 4a, b). Additional interactions between the portal and capsid are mediated by the wing domain's stem loop (Fig. 4a, b). The

tunnel loop from the wing domain narrows the portal channel of SU10 to 33 Å (Fig. 4, Supplementary Fig. 5). In phages T7 and P23-45, it was shown that the tunnel loops open the portal channel and thus regulate genome release[15,30]. However, the tunnel loops in the SU10 virion and genome release intermediate have the same structure, therefore it is likely that a different mechanism ensures the genome retention of SU10 (Supplementary Fig. 5). The stem domain connects the wing and clip domains and spans across the capsid shell (Fig. 4b). The clip domain of the portal complex reaches out from the capsid and enables the attachment of the adaptor proteins (Fig. 4a).

## Adaptor complex

The dodecamer of SU10 adaptor proteins forms the interface between the dodecamer of portal proteins and hexamer of nozzle proteins (Figs. 1c, 2a, and 5). The adaptor protein of SU10 has the same domain organization as those of phages T7 and KP32: a helix bundle domain, long tail fiber dock domain, and embracing loop (Fig. 5c)[15]. An embracing loop of adaptor protein is wedged between the clip

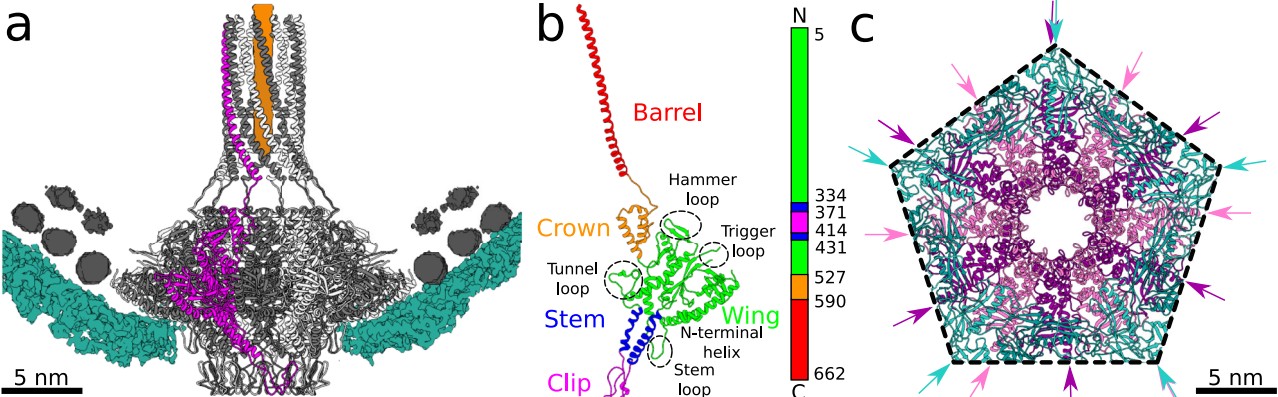

**Fig. 4 | Structure of SU10 portal complex and its interactions with capsid.**
**a** Cartoon representation of portal complex embedded in capsid. One portal protein is shown in magenta, and the remaining subunits in alternating white and grey. The cryo-EM density of the capsid is shown in turquoise, and that of DNA is shown in grey. The density of a putative end of the dsDNA genome or ejection proteins inside the barrel of the portal complex is shown in orange. The maps of capsid and DNA are shown at 2σ and 4σ, respectively. **b** Cartoon representation of a subunit of portal protein colored according to the domain composition. The domain

organization of portal protein is shown as a 1D plot. **c** Interactions of portal complex with capsid, viewed along tail axis from outside the particle. Portal proteins are shown in alternating pink and purple, capsid proteins are shown in dark and light turquoise. The directions of spine helices from the peripheral domains of major capsid proteins and N-terminal helices from the wing domains of portal proteins are highlighted with arrows to emphasize the mismatch between the fivefold symmetry of the capsid and twelvefold symmetry of the portal.

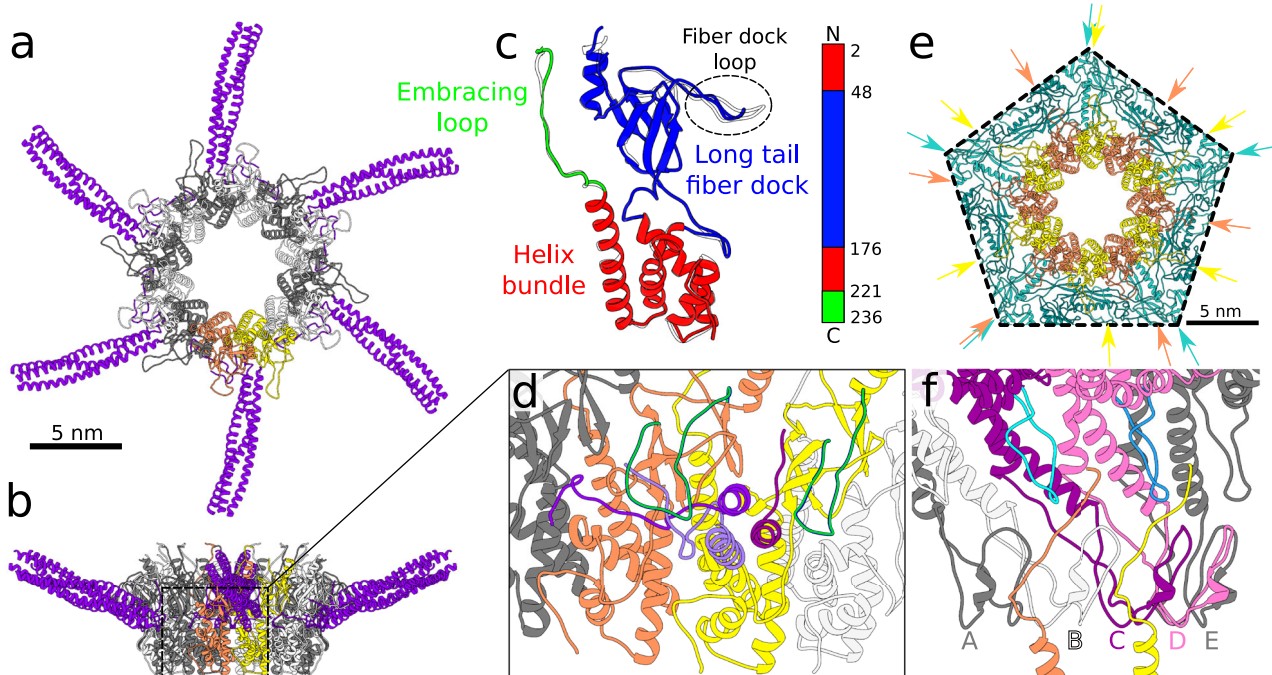

**Fig. 5 | Adaptor complex and its interactions with long tail fibers, portal, and capsid. a, b** Cartoon representation of adaptor complex decorated with six long tail fibers viewed along and perpendicular to tail axis. Two selected adaptor proteins from the dodecamer are highlighted in orange and yellow. The remaining subunits are shown in alternating white and grey. Each of the six long tail fibers is formed by a trimer of gp12 proteins (violet). **c** Superposition of structures of two neighboring adaptor proteins that differ in the conformation of their long tail fiber dock loops. One of the adaptor proteins is colored according to the domain composition. The superimposed subunit is shown in white. The domain organization of adaptor protein is shown as a 1D plot. **d** Interactions of adaptor proteins with long tail fiber. Fiber dock loops (green) of two neighboring adaptor proteins interact with one long tail fiber. The three subunits that form the long tail fiber are differentiated by shades of violet. The N-terminus of each long tail fiber protein from one long tail fiber forms unique interactions with the adaptor complex. **e** Interactions of adaptor complex with capsid viewed along tail axis from outside the particle. Adaptor

proteins are shown in alternating yellow and orange, capsid proteins are shown in shades of turquoise. Directions of spine helices from the peripheral domains of major capsid proteins and fiber dock loops of adaptor proteins are indicated with arrows to emphasize the symmetry mismatch between the capsid and adaptor complex. **f** Interface between portal proteins and C-terminal embracing loops of adaptor proteins. The embracing loop (orange) goes in between the portal clip domains of portal subunit A (grey) and B (white), then it crosses the stem of portal subunit C (dark magenta) and reaches between the wing loop (cyan) of subunit C and stem of subunit D (pink). The neighboring adaptor proteins differ in the structures of their embracing loops. The embracing loop (yellow) of the adaptor protein with the other conformation passes in between the portal clip domains of portal subunits B (white) and C (dark magenta), then crosses the stem of portal subunit D (pink) and reaches between the wing loop (light blue) of subunit D (grey) and stem of subunit E (pink).

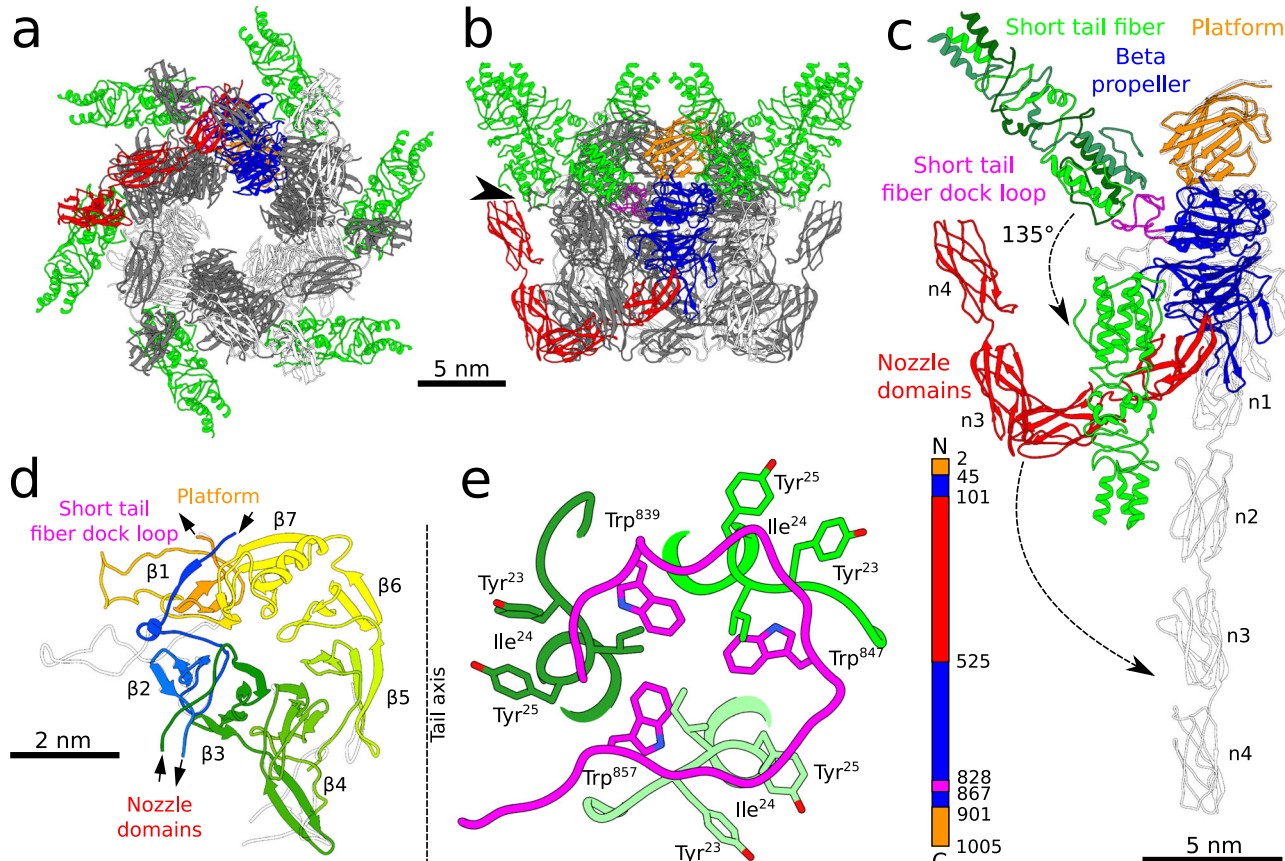

**Fig. 6 | Structure of nozzle proteins and short tail fibers.** Cartoon representation of hexamer of nozzle proteins with attached short tail fibers from SU10 virion viewed from outside the particle along (**a**) and perpendicular (**b**) to tail axis. The nozzle proteins are shown in alternating white and grey, and short tail fibers in green. One of the nozzle proteins is highlighted in color, with the platform domain shown in orange, beta-propeller domain in blue, short tail fiber dock in magenta, and four nozzle domains in red. The black arrowhead in panel (**b**) shows the interaction of nozzle domain 4 and the short tail fiber. **c** Comparison of structures of nozzle proteins and short tail fibers in SU10 virion and genome release intermediate. The nozzle protein from the virion is colored as in panels **a** and **b**, and the three gp16 subunits forming a short tail fiber are distinguished by shades of green. The superimposed structures of nozzle protein and short tail fiber from the genome release intermediate are shown in white and light green, respectively. The domain organization of nozzle protein is shown as a 1D plot. **d** Comparison of structures of beta-propeller domains from nozzle proteins of SU10 virion and genome release intermediate, shown in rainbow colors and white, respectively. The seven-bladed beta-propeller domain is rainbow-colored from the N-terminus in blue to the C-terminus in red. The short tail fiber dock loop is an extension of blade 1 ($\beta$1), whereas the nozzle domains are inserted between blade 2 ($\beta$2) and blade 3 ($\beta$3). **e** Interactions of short fiber dock loop (magenta) with three short tail fiber proteins (shades of green). The side chains of $Trp^{839}$, $Trp^{847}$, and $Trp^{857}$ from the short fiber dock loop form a star with quasi-threefold symmetry, which enables attachment of the short tail fiber, which is a trimer. Residues $Tyr^{23}$, $Ile^{24}$, and $Tyr^{25}$ from each short tail fiber protein clamp a segment of the fiber dock loop before each of the tryptophans.

domains of a pair of neighboring portal proteins and binds to stem loops of two additional portal proteins positioned clockwise when looking at the portal complex from outside the capsid (Fig. 5f). The long tail fiber dock domains of two neighboring adaptor proteins differ in structure because they interact with different parts of one long tail fiber (Fig. 5a, b, d). The SU10 long tail fiber dock domain is similar to those of phages T7 and KP32 (Supplementary Fig. 6)[15,31], which have long tail fibers[13,32]. In contrast, the adaptor proteins of phages P22 and Sf6, which do not have long tail fibers, lack the long tail fiber dock domains (Supplementary Fig. 6)[12,27].

### Long tail fibers
Six long tail fibers are attached to the sides of the adaptor complex (Figs. 1a–c, 2a and 5a, b). The long tail fiber of SU10 is formed by trimers of long tail fiber proximal (gp12) and distal (gp13) proteins. One long tail fiber branches from the helix bundle domain of every second subunit of the adaptor complex (Figs. 2a, c and 5a, b). The N-termini of two long tail fiber proximal proteins from one fiber extend to cover the surface of the helix bundle domain of the neighboring adaptor protein located counterclockwise when looking

at the adaptor complex from outside the capsid (Fig. 5b, d). The N-terminus of the third long tail fiber proximal protein binds to the long tail fiber dock domain of the adaptor protein from which the fiber branches out (Fig. 5d). The long tail fibers form additional contacts with the fiber dock loops that protrude radially from the fiber dock domains of adaptor proteins (Fig. 5a, c, d). The coiled-coiled part of each long tail fiber is held by fiber dock loops from two neighboring adaptor proteins (Fig. 5a, d). The cryo-EM reconstruction of the tail enabled the building of 90 residues from the N-terminus of the 786-residue long tail fiber proximal protein (Figs. 2a, c and 5a, b). The remainder of the long tail fiber was not resolved, probably because of its flexibility. Electron micrographs of purified phages show that the long tail fiber is 700 Å long and contains one elbow joint, which divides the fiber into 300- and 400-Å-long segments (Supplementary Fig. 4a, b).

### Structure of SU10 nozzle protein
The hexamer of nozzle proteins, attached to the adaptor, is decorated with six short tail fibers, and its central channel is plugged with a tail needle (Figs. 1a–c, 2a, c and 6). According to the convention

established for phage T7[15], the nozzle protein of SU10 can be divided into the platform, beta-propeller, and nozzle domains (Fig. 6a–c). However, the nozzle proteins of SU10 and T7 exhibit no detectable sequence similarity.

The platform domain has a beta-sandwich fold with the two beta-sheets formed by five and three beta-strands, respectively (Fig. 6c). The platform domain of each nozzle protein interacts with the helix bundle domains of two adjacent adaptor proteins. The distinct interactions of the nozzle proteins only induce minimal differences in the structure of the neighboring adaptor proteins.

The seven-bladed beta-propeller domain has a diameter of 50 Å and forms the core of the nozzle protein (Fig. 6d). Blades 4, 5, and 6 of the beta-propeller domain line the tail channel, whereas blades 1, 2, and 3 are distant from the tail axis. The beta-propeller domain is interrupted by two entry and exit points of the polypeptide main chain, so-called openings. The first opening in blade 1 includes the entry of the main chain from the N-terminal part of the platform domain and the exit of the main chain to the C-terminal part of the platform domain (Fig. 6d). The structure of blade 1 of the beta-propeller is stabilized by the antiparallel interaction of the outermost beta-strand, which is formed by C-terminal residues of the domain, with the remaining strands formed by residues from the N-terminus of the domain, a so-called velcro closure (Fig. 6d)[33]. The second opening is caused by the insertion of nozzle domains between beta-propeller blades 2 and 3 (Fig. 6d). Loops protruding from the edges of blades 4 and 5 enable the binding of the tail needle. The loops have different conformations in the genome release intermediate, resulting in the broadening of the tail channel diameter from 25 to 31 Å (Supplementary Fig. 5). The short tail fiber dock loop is located at the edge of blade 1 of the beta-propeller domain and enables the attachment of a short tail fiber (Fig. 6c–e).

In the SU10 virion, the Ig-like nozzle domains are organized into an L-shape arrangement (Figs. 2a, 6a–c). The first two nozzle domains form a blunt tip of the tail, then there is an 82° turn and the remaining two domains point towards the capsid (Fig. 6c). The contacts between nozzle proteins are mediated by the platform and beta-propeller domains. In addition, the nozzle domains wrap around the tail, and nozzle domain 1 interacts with nozzle domains 2 and 3 from the nozzle protein located clockwise when looking at the tail along its axis towards the capsid (Fig. 6a, b). Nozzle domain 4 interacts with the short tail fiber dock loop from the nozzle protein located counter-clockwise (Fig. 6a, b). The multiple interactions of nozzle domains stabilize the native structure of the SU10 tail.

## Short tail fibers

The short tail fiber of SU10 has a length of 210 Å, is straight, and can be divided into proximal, central, and receptor-binding domains (Figs. 2, 6a–c, Supplementary Fig. 7). The proximal domain was reconstructed to a resolution of 3.8 Å, which enabled the building of the first 98 residues of gp16 (Fig. 6a–c, Supplementary Fig. 7). The alpha helices that form the proximal domain are interrupted by short loops that mediate the swapping of positions of the helices from individual subunits within the domain (Supplementary Fig. 7). The central and receptor-binding domains modeled using the program AlphaFold2 multimer could be fitted into the 7 and 15 Å resolution cryo-EM reconstructions, respectively, with a real-space correlation coefficient of 0.81 (Supplementary Fig. 7A)[34]. The central domain is similar to the tip of the short tail fiber of phage T4 (PDB: 1PDI) (Supplementary Fig. 7)[35,36], whereas the receptor-binding domain resembles the tip of the long tail fiber of phage T4 (PDB: 2XGF), which recognizes lipopolysaccharides or the outer membrane porin protein C (Supplementary Fig. 7)[36]. Additionally, both the central and receptor-binding domains are similar to the corresponding parts of the receptor-binding protein of temperate bacteriophage JUB59 (PDB: 6OV6) (Supplementary Fig. 7)[37]. The SU10 short tail fiber protein contains two pairs of

histidines (His[194] and His[196], His[230] and His[232]), which may bind Fe²⁺ ions and thus stabilise the fiber, as occurs in T4 and JUB59[36–39].

The short tail fiber is attached to a fiber dock loop from a nozzle protein (Fig. 6c, e). The fiber dock loop contains tryptophans 839, 847, and 857, the side chains of which are arranged in a pseudo-threefold symmetry (Fig. 6e). Each tryptophan side chain forms hydrophobic interactions with Ile24 from one of the three short tail fiber proteins (Fig. 6e). Furthermore, the side chains of Tyr[23], Ile[24], and Tyr[25] of each short tail fiber clamp a segment of the polypeptide before the tryptophans of the fiber dock loop (Fig. 6e). Residues 3–6 of the short tail fiber interact with residues 303–308 from nozzle domain 4 of the nozzle protein located clockwise when looking along the tail axis towards the capsid (Fig. 6b). The position of the short tail fiber is additionally stabilized by the interaction of the central domain of the short tail fiber with residues 71–78 of the long tail fiber (Figs. 1c, 2a, c). It is likely that the disruption of these interactions enables coordination of the conformational changes of the long and short tail fibers, as well as those of the nozzle proteins, upon the binding of SU10 to the host cell.

## Tail needle

The structure of the SU10 tail needle with imposed threefold symmetry was reconstructed to a resolution of 15 Å (Figs. 1a–c, 2a, c, Supplementary Fig. 8). The structure of the tail needle, predicted by Alphafold2 multimer, can be divided into coiled-coil and C-terminal domains, and could be fitted into the cryo-EM density with a real-space correlation coefficient of 0.80 (Supplementary Fig. 8A, B). The coiled-coil domain of the SU10 tail needle is homologous to the tail needle of the phage P22[14], which was shown to bend up to 18°[40]. Therefore, we expect the tail needle of SU10 to be similarly flexible, which may be the reason why we could not resolve it to a high resolution. The C-terminal domain has a beta-prism structure similar to that of the central tail spikes of *Myoviridae* phages (Fig. 2a, c, Supplementary Fig. 8A, B)[38,41,42]. The binding of the SU10 tail needle to the tail channel is enabled by two loops on blade 4 and one loop on blade 5 of the beta-propeller domain of the nozzle proteins (Supplementary Fig. 8C, D). The 260-Å-long tail needle protrudes 200 Å from the nozzle complex (Figs. 1c, 2a, c).

## Head of SU10 genome release intermediate

The capsid of the genome release intermediate of SU10 is identical to that of the virion (Fig. 1, Supplementary Table 3). The induction of SU10 genome release in vitro by osmotic shock results in the incomplete ejection of the phage DNA (Supplementary Fig. 4). The reconstruction of the head of the SU10 genome release intermediate contains rings of putative DNA density attached to the inner face of the capsid (Fig. 1e, Supplementary Fig. 9). Even though the DNA density in the reconstruction of the genome release intermediate is organized in rings, the DNA remaining inside the head is most likely a single continuous strand. The rings are artefacts caused by the combination of structural variability of the DNA segments in individual particles and their misalignment during the reconstruction process. The propensity of the SU10 capsid to organize DNA strands, as observed in the genome release intermediate, is likely to influence the genome structure during packaging.

The asymmetric reconstruction of the capsid cap without a tail contains a spherical shell of density with an outer diameter of 80 Å and wall thickness of 15 Å (Supplementary Fig. 9C,E). This sphere of density is also visible in two-dimensional class averages of the genome release intermediates (Supplementary Fig. 9F). The sphere is not centered on the fivefold symmetry axis of the capsid, but is connected by a density to the beta-barrel formed by annular loops of major capsid proteins from a pentamer (Supplementary Fig. 9E). The position of the spherical shell of density leaves one edge of the pentamer accessible for binding DNA (Supplementary Fig. 9C). The asymmetric reconstruction of the SU10 virion did not contain this spherical shell of density, however, it is

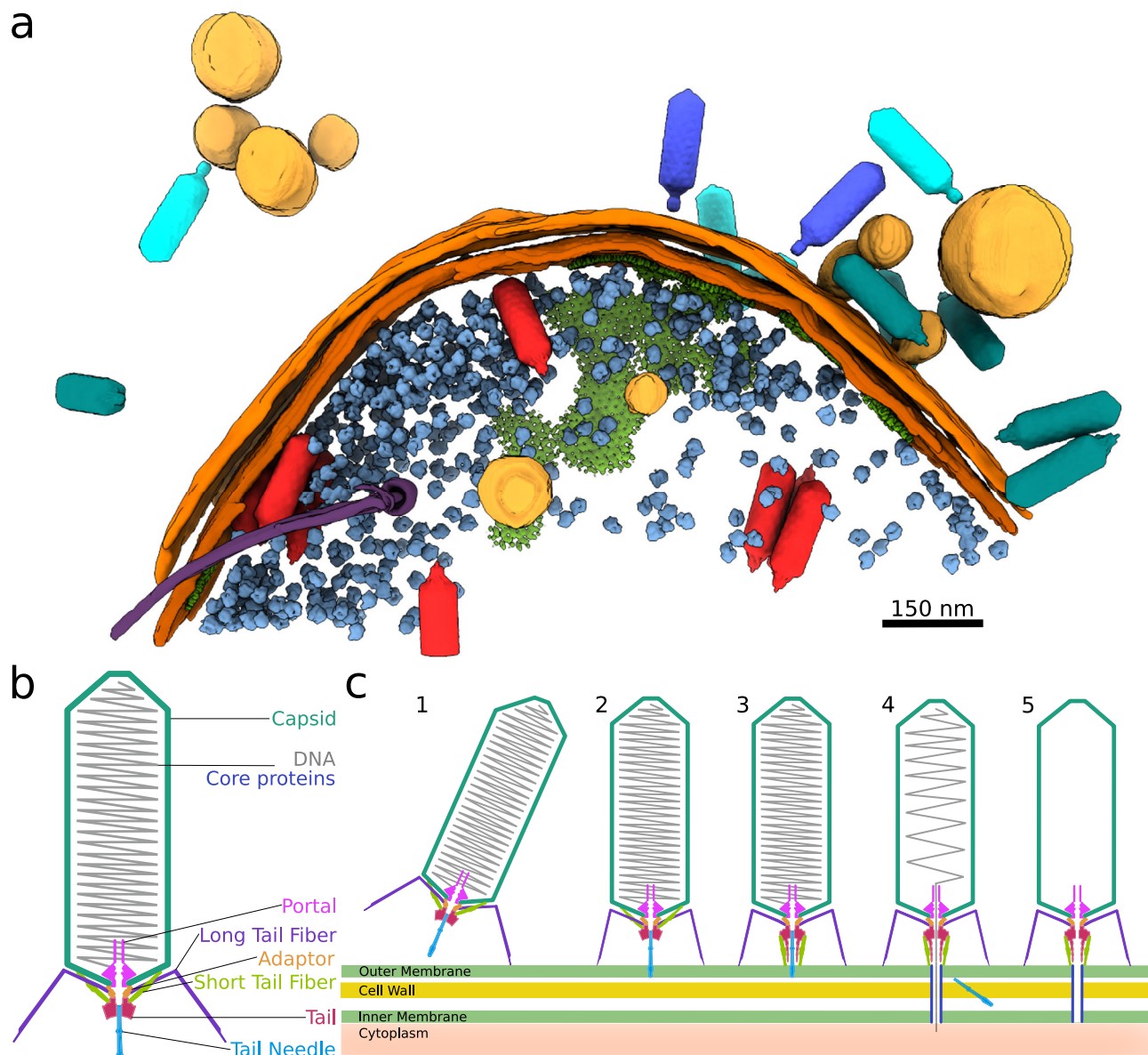

**Fig. 7 | SU10 structure and mechanism of genome delivery. a** Segment of infected *E. coli* cell 45 min post-infection contains SU10 phages at various stages of infection cycle. Virions outside the cell are shown in turquoise, genome release intermediates with formed nozzles in dark blue, empty particles in cyan, and progeny virions inside the cell in red. Inner and outer cell membranes are highlighted in dark orange and orange respectively, vesicles in light orange, ribosomes in light blue, chemoreceptor array in green, and flagellum in violet. Positions of phage particles, ribosomes (EMD-13270), and the chemoreceptor array (EMD-10160) were identified using template matching as implemented in EMclarity[64,65], and the corresponding high-resolution structures were positioned into the tomogram.

**b** Scheme of SU10 virion. **c** Mechanism of SU10 attachment and genome delivery. (1) The SU10 virion attaches to a cell surface via long tail fibers. (2) The binding of all six long tail fibers positions the phage with its long axis perpendicular to the bacterial membrane. The tail needle can interact with the outer membrane. (3) The short tail fibers rotate towards the cell membrane and the nozzle proteins straighten to form a nozzle. (4) The tail needle dissociates from the tail. Ejection proteins are released from the phage head, degrade cell wall peptidoglycan, and extend the phage nozzle to form a channel across the periplasm. The ejection of SU10 DNA into bacterial cytoplasm is initiated. (5) The empty particle remains associated with the cell.

possible that the asymmetric alignment required to resolve the sphere structure was prevented by the strong signal from the genome. It is not clear what forms the observed spherical shell of density. The radius of the sphere is too small for it to be formed by bent dsDNA[43]. Therefore, it is likely that the spherical shell is formed by proteins.

**Structural rearrangement of SU10 tail for genome release**

The tail of the SU10 genome release intermediate is reorganized into the nozzle built from nozzle proteins and short tail fibers (Figs. 2c,d, 6c, Supplementary Fig. 5, Supplementary Movie 1). To form the nozzle, the four nozzle domains undergo an extensive structural change from

the L shape in the SU10 virion to the straight arrangement in the genome release intermediate (Fig. 6c). In the new conformation, the four domains form a line pointing away from the portal complex. While the interaction of the short tail fiber dock loop with a short tail fiber remains preserved, the connections of the short tail fiber loop to the beta-propeller domain rotate about each other and bend 135° to point away from the phage head (Fig. 6c, d). The same rotation is performed by the short tail fiber. In the new arrangement, the nozzle proteins and short tail fibers alternate to form a nozzle that prolongs the tail by 277 Å (Figs. 1d, 2b, d). Furthermore, the distal part of the beta-propeller domain of the nozzle protein shifts 3 Å away from the

tail axis, resulting in a broadening of the tail channel (Fig. 6c, Supplementary Fig. 8C, D). The formation of the nozzle is required to bring the receptor-binding domains of short tail fibers into a position in which they can interact with the outer membrane of *E. coli*.

## Cryo-ET and fluorescence microscopy of SU10-infected cells

Tomograms of SU10-infected *E. coli* demonstrate that multiple phages can bind and deliver their genomes into one cell (Fig. 7a, Supplementary Fig. 10, Supplementary Movie 2). Some of the phages were only attached by long tail fibers, whereas the tails of the remaining phages formed nozzles (Fig. 7a, Supplementary Fig. 10). Furthermore, the phages with nozzles were in various stages of genome release, from full to empty heads. The empty particles remained attached to the cell surface (Fig. 7a, Supplementary Fig. 10D–F). In the late stages of SU10 infection, the sample contained outer membrane vesicles with bound phage particles (Fig. 7a, Supplementary Fig. 10J–L). The membrane vesicles can be shed by bacteria growing in planktonic culture[44–46]. Under the conditions of *E. coli* cultivation used in our experiments, numerous progeny virions assembled in the infected cells 45 to 65 min post-infection (Supplementary Fig. 10J–L). The few initially formed particles were distributed randomly, however over time after infection, the progeny virions assembled into regular arrays, as has been described previously[2].

Airyscan confocal images of *E. coli* cells with labelled membranes and genomes, which were infected by SU10 with labelled DNA, verified that numerous phage particles bind and deliver their genomes into one cell (Supplementary Fig. 10C–F). The fluorescence signal from the labeled phage DNA was detectable in the cell cytoplasm until cell lysis, and it was even incorporated into progeny virions (Supplementary Fig. 10I, L). The infected cells retained their native shape unit 45 min post-infection, however at 65 min they lost turgor (Supplementary Fig. 10J–L).

## Mechanism of SU10 genome delivery

The initial binding of the SU10 virion to receptors in the outer membrane of *E. coli* is mediated by long tail fibers, which have receptor-binding domains at their distal ends (Fig. 7b, c). The structures of the SU10 virion and genome release intermediate differ in the direction in which the long tail fibers point away from the adaptor complexes (Fig. 2c, d). The long tail fibers in the genome release intermediate are rotated 6.6° relative to their orientation in the virion. We speculate that under native conditions this conformational change is induced by phage binding to the bacterial membrane. Furthermore, the movement of the long tail fibers probably disrupts their interactions with short tail fibers and thus primes them for the formation of the tail nozzle. The binding of all six SU10 long tail fibers to receptors positions the phage with its long axis perpendicular to the bacterial membrane (Fig. 7a, c). The putative interaction of the tail needle with the bacterial membrane triggers conformational changes to nozzle proteins, which lead to the disruption of the interaction of nozzle domain four with the short tail fiber. By arranging into a straight line parallel with the tail axis, the nozzle domains make room for the 135° rotation of the short tail fibers to point away from the phage head. The short tail fibers and nozzle proteins alternate to form a 277-Å-long nozzle (Fig. 7c). The formation of the nozzle enable the predicted receptor-binding domains of short tail fibers to interact with receptors at the cell surface. The binding of the short tail fibers to receptors may force the tail needle, which protrudes from the nozzle, through the outer *E. coli* membrane. Subsequently, the tail needle dissociates from the tail to enable the opening of the tail channel (Fig. 7c). Ejection proteins are probably ejected from the SU10 head before or together with the beginning of the genome. The ejection proteins form a translocation channel across the periplasmic space and the ejection protein gp20, with transglycosylase activity, degrades the cell wall peptidoglycan. The phage genome is delivered to the *E. coli* cytoplasm

through a channel that is formed by the nozzle and the translocation channel (Fig. 7c).

# Methods

## SU10 propagation and purification

Phage SU10 was propagated on *E.coli* strain ECOR10, grown at 37 °C in nutrient broth (Oxoid). Phage lysate from 300 ml of bacterial culture was treated with turbonuclease (Abnova) (final concentration 5 Units/ml), centrifuged at 6000 x *g* for 20 min at 4 °C, and filtered through a 0.45 µm polyethersulfone syringe filter. Phages from the filtrate were pelleted by centrifugation at 54,000 x *g* for 2.5 h at 4 °C in a 50.2 Ti rotor (Beckman Coulter). Phage pellets were dissolved by overnight incubation in 6 ml of a phage buffer (50 mM Tris-Cl, 10 mM NaCl, 10 mM CaCl$_2$, pH 8.0) at 4 °C. Dissolved pellets were overlaid onto a step CsCl density gradient (3 ml of each CsCl solution in phage buffer with densities 1.45 g/ml, 1.50 g/ml, and 1.70 g/ml) and centrifuged at 210,000 x *g* for 4 h at 12 °C using an SW40Ti rotor (Beckman Coulter). Bands containing phage particles were collected using a 0.8 mm gauge needle and syringe. Caesium chloride was removed by dialysis against the phage buffer at 4 °C overnight using Visking dialysis tubing type 8/32″, 0.05 mm thick (part no. 1780.1, Carl Roth, Germany).

## Induction of phage genome release

The purified phage sample ($10^{12}$ PFU/ml) was 10 x diluted using a solution of 3 M urea in the phage buffer and incubated at 42° for 30 min. After incubation, the sample was 3x diluted in the phage buffer and treated with Turbonuclease (Abnova) at a final concentration of 3.5 Units/ml for 15 min at room temperature. Phage particles were pelleted by centrifugation at 75,000 x *g* at 4° for 1 h in a 50.2 Ti rotor (Beckman Coulter) and resuspended in the phage buffer. The protein composition of SU10 virions and genome release intermediates were compared using SDS gel electrophoresis and mass spectrometry analysis.

## Identification of structural proteins of phage SU10

The purified SU10 phage was resuspended in Laemmli buffer and boiled for 3 min, then the sample was treated with Turbonuclease at a final concentration of 2 Units/ml (Abnova) for 15 min at room temperature. Proteins were separated by tricine gradient gel electrophoresis. All major protein bands were cut from the gel and used for mass spectrometry analysis. Mass spectrometry data processing and analyses were performed using the software FlexAnalysis 3.4 and MS BioTools (Bruker Daltonics). Mascot software (Matrix Science, London, UK) was used for sequence searches in exported MS/MS spectra against the National Center for Biotechnology Information database and a local database supplied with the expected protein sequences. The mass tolerance of peptides and MS/MS fragments for MS/MS ion searches were 50 parts per million and 0.5 Da, respectively. The oxidation of methionine and propionyl-amidation of cysteine as optional modifications and one enzyme miscleavage were set for all searches. Peptides with a statistically significant peptide score ($P < 0.05$) were considered.

## Cryo-EM sample preparation, data collection, motion correction, and CTF estimation

Purified phage solution (4 µl) with a concentration of $10^{12}$ PFU/ml was pipetted onto a holey carbon-coated copper grid (R2/1, mesh 200; Quantifoil), blotted and vitrified by plunging into liquid ethane using an FEI Vitrobot Mark IV. The vitrified sample was transferred to a Thermo Fisher Scientific Titan Krios electron microscope operated under cryogenic conditions and at an acceleration voltage of 300 kV. The beam was aligned for parallel illumination in NanoProbe mode, and coma-free alignments were performed to remove the residual beam tilt. Micrographs of SU10 virions were collected at a magnification of 59,000x on a Falcon III direct electron detector operating in

linear mode, resulting in a calibrated pixel size of 1.38 Å/pix. Imaging was done under low-dose conditions (total dose 49 e⁻/Å²) and defocus values ranging from −1.2 to −2.7 μm. The one-second exposure was fractionated into 40 frames and saved as a movie. The dataset of genome release intermediates was collected at a magnification of 105,000 x at a K2 Summit direct electron detector operating in counting mode, resulting in a calibrated pixel size of 1.34 Å/pix. Imaging was done under low-dose conditions (total dose 52 e⁻/Å²) and defocus values ranging from −1.2 to −2.7 μm. The eight-second exposure was fractionated into 40 frames and saved as a movie. Automated data acquisitions were performed using the software EPU (Falcon III detector) or Serial EM (K2 Summit detector). The movies were motion-corrected globally and locally (5 × 5 patches) using the software MotionCor2[47] and saved as dose-weighted micrographs. Defocus values were estimated from aligned non-dose-weighted micrographs using the program Gctf[48].

## Cryo-ET of SU10 virions

Grids of SU10 virions prepared for single-particle acquisition were used to record tomography tilt series covering an angular range from −60° to +60° with 2° increments, with the bidirectional tilting scheme under low-dose conditions using SerialEM. Each tilt series was collected with a nominal defocus value of −6 μm. Each image was acquired as a movie of 7 frames at a Falcon II detector operated in counting mode using a dose of 1 e⁻/Å² per tilt. The total cumulative dose for each tilt series was 57 e⁻/Å². The calibrated pixel size was 1.8 Å. Tomograms were reconstructed using eTomo from the package IMOD[49]. Subtomograms of heads of SU10 virions were averaged using the program PEET[50,51] from the package IMOD with applied D5 symmetry.

## Reconstruction of head of SU10 virion

CrYOLO from the software package SPHIRE neural network was trained on a subset of images of heads of SU10 virions picked manually[52]. Final auto picking was done on 4 x binned micrographs using crYOLO. The resulting particle coordinates were corrected for the binning factor, and images of particles (box size 1296 × 1296 pixels) were extracted from the 9,000 original unbinned micrographs using Relion[53]. Particles were binned to 256 px (6.99 Å/pixel) using reciprocal space binning as implemented in the software package Xmipp[54]. Several rounds of 2D classification were performed in Relion to select intact virions from the auto-picked set of particles. The initial model of the SU10 head from subtomogram averaging was low-pass filtered to a resolution of 30 Å. Several rounds of 3D classification and refinement with imposed C5 symmetry were performed using Relion 3.1. The software package Spider[55] was used to prepare a cylindrical mask excluding the phage genome to prevent the signal from the genome from interfering with the capsid reconstruction. After initial auto refinement with imposed C5 symmetry, 3D classification was performed without the alignment step. Particles belonging to the best class were selected for further Relion auto refinement with imposed C5 symmetry and maximum allowed deviations from previous orientations of 10°. As the resolution of the reconstruction increased, the particles were gradually unbinned to 512 px (3.49 Å/pixel) and 1296 px (1.38 Å/pixel). The final C5 symmetrized reconstruction was threshold-masked, divided by the modulation transfer function, and B-factor sharpened during the post-processing in Relion 3.1. To further improve the resolution of the capsid, we divided the capsid into the C5 symmetric cap without a tail, D10 symmetric central part, and C5 symmetric cap with a tail. All these parts were re-extracted into smaller boxes, and their reconstructions were refined using local angular searches. To obtain an asymmetric reconstruction of the phage tail and the surrounding capsid, symmetry-relaxation from C5 to C1 was performed using the program relion_relax_symm (Supplementary Fig. 11).

## Reconstructions of portal and tail from SU10 virion

Particle orientations from the C5 reconstruction of the phage head were used to extract sub particle images centered on portals and tails of SU10 virions. The images of sub-particles were subjected to several rounds of 2D classifications, which enabled the selection of homogeneous datasets. After initial auto refinement with imposed C12 symmetry for the portal and C6 and C3 symmetries for the tail, 3D classifications were performed without the alignment step. Particles belonging to the best classes were selected for further Relion auto refinement with imposed symmetries and maximum allowed deviations from previous orientations of 10°. The resulting maps were threshold-masked, divided by the modulation transfer function, and B-factor sharpened during the post-processing in Relion (Supplementary Fig. 11).

## Tail needle reconstruction

Particle orientations from the C6 reconstruction of the phage tail were used to extract sub-particle images centered on the tail needle of the SU10 virion. The images of sub-particles were subjected to several rounds of 2D classifications, which enabled the selection of a homogeneous dataset. After initial auto-refinement with relax symmetry set to C3, 3D classification was performed without the alignment step. Particles belonging to the best class were selected for further Relion auto-refinement with imposed C3 symmetry and maximum allowed deviations from previous orientations of 10° (Supplementary Fig. 11).

## Asymmetric reconstruction of SU10 virion

To obtain an asymmetric reconstruction of the whole SU10 virion, the orientations from the C5 symmetrized head reconstruction were relaxed using relion_relax_symm. The software package Spider[55] was used to prepare a spherical mask covering the phage tail and 40 nm of the capsid.

The initial reconstruction was performed on particle images binned to 256 px (6.99 Å/pixel), using a high regularization factor and allowing only rotational searches. The successful relaxation of the symmetry was verified by the presence of biologically relevant features in the parts of the reconstruction belonging to both the head and tail of the phage. Subsequently, the reconstructions were continued with data binned to 512 px (3.49 Å/pixel) and 1296 px (1.38 Å/pixel) (Supplementary Fig. 11).

## Reconstruction of SU10 genome release intermediate capsid

SU10 genome release intermediates were manually boxed using e2boxer from the software package EMAN2[56]. The resulting particle coordinates were used to extract particles from the 5688 micrographs using Relion[53,54]. Particles were binned to 256 px (6.99 Å/pixel) using reciprocal space binning as implemented in the software package Xmipp[54]. Several rounds of 2D classification in Relion were performed to select intact genome release intermediates from the set of particles. The reconstruction of the capsid of the SU10 virion, low-pass filtered to a resolution of 30 Å, was used as the initial model for 3D reconstruction in Relion. Several rounds of 3D classification and refinement with imposed C5 symmetry were performed using Relion 3.1. The resulting map was threshold-masked, divided by the modulation transfer function, and B-factor sharpened during the post-processing in Relion (Supplementary Fig. 12).

## Reconstruction of tail of SU10 genome release intermediate

Particle orientations from the C5 reconstruction of the capsid of SU10 genome release intermediate were used to extract sub-particle images centered on tails. The images of sub-particles were subjected to several rounds of 2D classifications, which enabled the selection of a homogeneous dataset. After initial autorefinement with imposed C6 symmetry, 3D classification was performed without the alignment step. Particles belonging to the best class were selected for further

Relion autorefinement with imposed C6 symmetry and maximum allowed deviations from previous orientations of 10°. The resulting map was threshold-masked, divided by the modulation transfer function, and B-factor sharpened during the post-processing in Relion (Supplementary Fig. 12).

### Asymmetric reconstruction of capsid top of SU10 genome release intermediate

Particle orientations from the C5 reconstruction of the capsid of the SU10 genome release intermediate were used to extract sub-particle images centered on capsid tops. The images of sub-particles were subjected to several rounds of 2D classifications, which enabled the selection of a homogeneous dataset. Particles were C5 symmetry expanded, and 3D classification into ten classes was performed without the alignment step. Particles belonging to the best classes were selected, and particle duplicates from symmetry expansion were removed. The resulting particles were used for further Relion autorefinement without symmetry and with maximum allowed deviations from previous orientations of 10° (Supplementary Fig. 12).

### Cryo-EM structure determination and refinement

PDB structures were built using the program COOT[47] (portal protein, adaptor protein, residues 1-90 of long tail fiber, and proximal domain of short tail fiber), built automatically using Deep Tracer[57] (nozzle protein) or modeled using Alphafold2[34] (central and receptor-binding domains of short tail fiber and tail needle). The structures were iteratively refined in real space using the program PHENIX real_space_refine.py[58] and corrected manually in COOT[47] and ISOLDE[59]. The quality of the structures and their fit to the cryo-EM density maps was assessed using comprehensive validation in the program PHENIX. FSC curves of cryo-EM reconstructions are presented in Supplementary Fig. 13. Model to map FSC curves of individual structures, and examples of the fit of PDB models to cryo-EM densities are shown in Supplementary Fig. 14.

### AlphaFold2 structure prediction

Structures of SU10 proteins that were not resolved in cryo-EM reconstructions to sufficient resolution to enable structural determination were predicted using a local installation of AlphaFold multimer 2.1.2. The software was installed from https://github.com/deepmind/alphafold, using a Dockerised environment. For prediction, databases recommended by AlphaFold authors at the time of installation (winter 2021), were used (BFD, MGnify, PDB70, PDB, PDB seqres, Uniclust30, UniProt, UniRef90)[34].

### Preparation of composite cryo-EM maps of SU10 virion and genome release intermediate

The composite map of the SU10 virion was prepared by combining segments from sub particle reconstructions of SU10 virion regions. Sub-particle reconstructions of regions of the SU10 virion were fitted into asymmetric reconstruction of the complete SU10 virion using Chimera[60–62] (the Fit in Map command). The composite map was built from the following reconstructions: C5 symmetrized capsid reconstruction; portal from C12 symmetrized neck reconstruction; adaptor, long tail fibers, nozzle, and short tail fibers from C6 symmetrized tail reconstruction; and C3 symmetrised tail needle reconstruction. Segments of sub-particle reconstructions were combined into the composite map using the following procedure: (1) Masks covering PDB structures of the component proteins were generated using pdb2mrc. (2) The masks were extended by six voxels and an additional four voxels of soft edge (Relion3 - relion_mask_create). (3) The sub-particle reconstructions were multiplied by the corresponding masks (Relion3 - relion_image_handler). (4) For each segment threshold the structure of the entire component with maximum resolved details was selected for display. For a short tail fiber map with decreasing resolution along the fiber, three segments with different thresholds were selected. (5)

The maps of segments were normalized using CCP4 mapmask. (6) The normalized segments were combined into the final composite map using the Chimera command vop add onGrid. The composite map of the SU10 genome release intermediate was prepared using the same procedure from capsid reconstructed with C5 symmetry and tail reconstructed with C6 symmetry.

### Fluorescence microscopy sample preparation, data collection, and processing

An exponentially growing culture of *E. coli* ECOR10 (OD = 0.3) was infected with bacteriophage SU10 (MOI = 10) pre-stained with DmaO (BIOTIUM) in the presence of 0.002 M $CaCl_2$. The infection was performed in a PS 75/25 mm cellview cell culture slide with a glass bottom (Greiner bio-one) at 37 °C, with 100 RPM shaking and allowed to proceed for 5, 25, 45, and 65 min. Bacterial membranes and DNA were stained with SynaptoRed (BIOTIUM) (final concentration 4 μM) and DAPI (Roche) (final concentration 300 nM). For imaging, the samples were fixed with a mixture of glutaraldehyde and formaldehyde (0.125% and 4%, respectively) in a 50 mM phosphate buffer (pH=7.5). To prevent motion during imaging, samples were overlaid with 1% low-melting agarose in the phage buffer.

Samples were imaged using a Zeiss LSM 880 confocal microscope with Airyscan in R-S mode, with fluorophore excitation at the following wavelengths: DAPI at 405 nm, DmaO at 488 nm, and SynaptoRed at 514 nm. Data was processed using ZEN Black and visualized using Fiji[63].

### Cryo-ET of SU10-infected *E. coli* cells

An exponentially growing culture of *E. coli* ECOR10 (OD = 0.3) was infected with the bacteriophage SU10 (MOI = 10). The infection was allowed to progress for 5, 25, 45, and 65 min. 1.5 ml of infected bacterial culture was pelleted by centrifugation at 10,000 x *g* for 1 min at 4°. The pellets were resuspended in 15 μl of the phage buffer and kept on ice. Gold fiducial markers (10 nm beads) were applied to the holey carbon-coated copper grid (R3.5/1, mesh 200; Quantifoil) for electron microscopy, wicked with filter paper and immediately covered with 4 μl of cell suspension. Subsequently, the grids were blotted and vitrified by plunging into liquid ethane using a Vitrobot Mark IV. Tomographic tilt series covering an angular range from −60° to +60° with 3° increments were recorded automatically using a dose-symmetric tilting scheme under low-dose conditions at a K2 Summit direct electron detector, positioned behind the energy filter in the Titan Krios microscope. Each tilt series was collected with a nominal defocus value of −6 μm and acquired as a movie containing 5 frames in counting mode of the detector, using a dose of 2 e⁻ /Å² per tilt. The total cumulative dose for each tilt series was 57 e⁻ /Å². The calibrated pixel size was 4.33 Å. Tomograms were reconstructed and visualized using IMOD from the software package eTomo[58]. Membranes and flagella were segmented using the program Amira[64]. Positions of phage particles, ribosomes (EMD-13270), and the chemoreceptor array (EMD-10160) were identified using template matching as implemented in EMclarity[64,65], and the corresponding high-resolution structures were positioned into the tomogram using in-house scripts https://github.com/fuzikt/tomostarpy.

### Reporting summary

Further information on research design is available in the Nature Research Reporting Summary linked to this article.

## Data availability

The cryo-EM reconstructions and corresponding PDB structures were deposited under the following EMDB and PDB codes: Structures of SU10 virion: capsid with fivefold symmetry EMD-14488 and PDB-7Z49, asymmetric capsid reconstruction EMD-14492 and PDB-7Z4B, capsid top EMD-14485 and PDB-7Z46, capsid center EMD-14484 and PDB-7Z45, capsid bottom EMD-14487 and PDB-7Z48, assymetric

reconstruction of capsid bottom and tail EMD-14489 and PDB-7Z4A, neck with twelvefold symmetry EMD-14483 and PDB-7Z44, tail with sixfold symmetry EMD-14486 and PDB-7Z47, tail needle with threefold symmetry EMD-14909, and composite map of the whole virion EMD-14977. Structures of genome release intermediate: capsid with fivefold symmetry EMD-14490, tail with sixfold symmetry EMD-14495 and PDB-7Z4F, capsid top EMD-14491, and composite map of the whole genome release intermediate EMD-14920.

## Code availability
Scripts for positioning high-resolution structures into the tomograms are available from https://github.com/fuzikt/tomostarpy.

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

## Acknowledgements

We gratefully acknowledge CEITEC MU Cryo-electron microscopy and tomography core facility and Proteomics Core Facility of CIISB, Instruct-CZ Centre supported by MEYS CR (LM2018127) for their assistance with obtaining the scientific data presented here. We acknowledge the core facility CELLIM supported by the Czech-BioImaging large RI project (LM2018129 funded by MEYS CR) for their support with obtaining scientific data presented in this paper. We greatly appreciate access to computing and storage facilities owned by parties and projects contributing to the National Grid Infrastructure MetaCentrum, provided under the program Projects of Large Infrastructure for Research, Development, and Innovations (LM2010005). This study was supported by the IT4Innovations Centre of Excellence (project CZ.1.05/ 1.1.00/02.0070). This work, including the efforts of Pavel Plevka, was funded by Czech Ministry of Education, Youth and Sports ERC-CZ Consolidator grant LL1906, Czech Science Foundation grant GA18-17810S, and the project National Institute of Virology and Bacteriology (Program EXCELES, ID Project No. LX22NPO5103) - Funded by the European Union - Next Generation EU. The funders had no role in the design of the study, data collection and interpretation, or the decision to submit the work for publication.

## Author contributions

Conceptualization P.P.; methodology M.Š., T.F., M.P., J.N., and M.B.; validation M.Š. and T.F.; formal analysis M.Š., M.P., J.N, and M.B.; investigation M.Š., M.P., J.N., and M.B.; resources A.N.; data curation M.Š. and T.F.; writing—original draft preparation M.Š., T.F., and P.P.; writing—review and editing M.Š., T.F., and P.P.; visualization M.Š. and T.F.; supervision P.P.; funding acquisition P.P.

## Competing interests

The authors declare no competing interests.
