## [Peer Review File · Nature Communications]

Reviewers' comments:

Reviewer #1 (Remarks to the Author):

In the MS "Tail proteins of Podoviridae phage SU10 reorganize into the nozzle for genome delivery" Siborova et al describe the structure of a podophage with a very long (very prolate) capsid and short tail called SU10. The structure of the virus particle has been investigated in its pre-infection, head-packaged state and post-infection, head-empty (nearly empty) state. A tomography of phages in the midst of infection and fluorescence imaging of the infection process has been also presented. Various parts of the virus particle have been reconstructed using cryo-EM and appropriate symmetries.

Overall, this is a big (very big) body of solid work.

However, I am very concerned about the presentation of the actual results.

Consider Fig. 1. The figure legend of panels A and B (it should be A, B here and throughout, not AB), reads

"Surface representation of cryo-EM reconstruction of virion of phage SU10 colored according to protein type..."

What I see is a _composite_ cryo-EM map, which consists of 6-7 cryo-EM maps, each obtained _separately_ using different symmetries. Somehow, these reconstructions are then put together to give a new "map", which is further segmented and rendered at some arbitrary contour levels. Neither the process of combining these maps together, nor the selection of the contour level for display of such complex objects, nor the process of segmentation and the removal "unwanted" parts is explained anywhere in the MS.

Now consider Fig.2. "Conformational changes of SU10 tail associated with genome release. "(AC) Cartoon representations of tail of virion (BD) and genome release intermediate of phage SU10". Again (B, D) and (A, C). What the figure fails to mention is that a large part of the shown material are Alphafold models. The cryo-EM map(s) was (were) too low resolution for model building.

Furthermore, consider the attached Validation Reports of the atomic models that were built and refined against cryo-EM maps (N.B. the "full portal" report is a damaged file). First of all, the reports have "This wwPDB validation report is NOT for manuscript review" across EVERY page. Secondly, the amount of bad residues - bad fit to the map or bad geometry - is pretty staggering. In some cases, 80% (!) of the atomic model is bad (e.g. in the Nozzle C6 report). Can these atomic structures be trusted at all?

Finally, when referring to Alphafold models, the authors never explain or show the predicted accuracy or quality of Alphafold models. In most figures, we do not know whether we are looking at an AF model or real structure. Fig. S9 is probably the only figure where the legend states that most of the ribbon diagram shown is an AF model. In a way, it would interesting to see the AF model for the whole protein.

I am not sure which of the two Alphafold multimer versions the authors used in their work, but this version:

ColabFold: AlphaFold2 using MMseqs2

<https://colab.research.google.com/github/sokrypton/ColabFold/blob/main/AlphaFold2.ipynb> despite the authors' claim of capable of predicting multimeric structures, failed miserably on all sequences I tried (produced absolute gibberish).

However, the official Alphafold Colab Pro version:

<https://colab.research.google.com/github/deepmind/alphafold/blob/main/notebooks/AlphaFold.ipynb> works well for moderately sized multimers. A locally installed version works even better.

If reported unaltered (without refinement against the cryo-EM map), AF models must be subject to

proper validation. A good correlation coefficient with the cryo-EM map tells us nothing about whether the model is real or not. The model can have clashes galore and fit the map perfectly.

Minor concern

The Cryo-ET and fluorescence microscopy section. I am not sure what are we learning in this section. Is this process different to that in other phages? Was there a hypothesis that needed to be tested? It is difficult to write this because, obviously, a lot of work went into producing these images, but I do not know if all this work add any new information to what we already know about phage infection.

There are several other comment/suggested edits aimed at making this MS better.

The abstract needs a major overhaul.

The structure of the phage particle must be described first. Briefly. The classification and phage therapy stuff can be removed, in principle.

L. 20-21

The binding of the long tail fibers to the receptors in the outer bacterial membrane is followed by the straightening of nozzle proteins and rotation of short tail fibers by 135°. ^[1]_{USEP}
We do not know what those things are at this point. Are they part of the phage particle or displayed on the host cell surface?

L. 22. Prolongs -> extends

L. 23. What is a tail needle? We do not know.

L. 24. What is "the inner core"? A core of what?

The last sentence should be somewhere in the middle or even earlier in the abstract. After the "we show" sentence, a more detailed description of the results should be given. Then, if there is room, the study can be placed into a wider context and phage therapy can be mentioned.

Results

The ring densities in the map of the empty (or partially empty) capsid shown Fig. S5 are an averaging artifact. In the raw micrograph shown in Fig. S4 we see rings all over the inside of the capsid. They are not separated into two different types. A presumably single strand of DNA can be see. Averaging split it into two types of rings. But neither appears to correspond to the original structure.

L. 206-247. I am not sure what new about portal proteins we are learning here from this very detailed description. Please shorten and concentrate on novel aspects.

L. 289-291 and Fig. S8. Comparison of structures of receptor-binding domains of long tail fibers of tailed phages. Well, we are looking at chaperones (!) of LTF in tailed phages. These chaperones fall off the tip of the fiber. The entry for the cited PDB 3GUD is called "Crystal structure of a novel intramolecular chaperon" (!).

Lots of things published about those intramolecular chaperones. The cleavage site is well defined and is very conserved. See, for example

<https://www.ncbi.nlm.nih.gov/pmc/articles/PMC2666599/>
<https://pubmed.ncbi.nlm.nih.gov/17158460/>
<https://pubmed.ncbi.nlm.nih.gov/19450535/>

Furthermore, the AF model shown in Fig. S8 panel A is clearly incorrect. The three beta-sheets at the bottom of the protein MUST form a beta-helix, similar to the structure shown in panel C (and other structures described in the references above and many others!).

So, the "thing" that protrudes upwards is a chaperone that is not present in the mature fiber. It does not interact with cell surface receptors.

L. 295-296 ...by a trimer of tail needle proteins.  I think in this context the needle protein is a singular entity that contains 3 polypeptide chains. Not three tail need proteins.

L. 292-338. How does the structure of this tail compares to other tails? Is a similar beta-propeller found elsewhere? What does sequence conservation mapped on the structure look like?

Fig. S6. Panel A. Why is only a part of the nozzle model shown?

L. 357- 359. The HxH motif is common in tail fibers and central spike proteins. See, e.g.

<https://journals.plos.org/plosone/article/comments?id=10.1371/journal.pone.0211432>

<https://pubmed.ncbi.nlm.nih.gov/26295253/>

[https://www.cell.com/fulltext/S0969-2126\(11\)00469-2](https://www.cell.com/fulltext/S0969-2126(11)00469-2)

<https://pubmed.ncbi.nlm.nih.gov/22922659/>

The HxH was reported to bind Zn in T4 gp12. This is likely an artifact of the gp12 purification procedure where the protein is partially denatured in the presence of EDTA first and then refolded in the presence of Zn ions.

So, it appears that all data shows that HxH motifs in trimeric fibers and tailspikes bind a Fe ion (Fe²⁺ to be more specific).

Fig. S9 is the only figure in which the figure legend states what is shown in the figure. Here in Panel A, and everywhere where a density is shown, the figure legend MUST state the contour level of the cryoEM map in standard deviations above the mean.

Fig. S10. The contour level is not stated (see Fig. S9 notes). Panel B is not an additional evidence that supports panel A. The model shown in panel A already contains the predicted secondary structure information (it has been built using that information). A much more informative panel would be one that shows the reliability of AF model - the IDDT values either mapped onto the structure or as a plot, and the Predicted Aligned Error plot. This is applicable to all AF models.

All ribbon diagram drawings must have residue numbers in strategic locations. E.g. we have no idea how to relate what's shown in panels A and B in Fig. S10.

Reviewer #2 (Remarks to the Author):

In the manuscript "Tail proteins of Podoviridae phage SU10 reorganize into the nozzle for genome delivery" by Šiborova et al, the authors use cryo-electron microscopy (both single particle analysis and tomography) to determine the structure of the phage. The cryo-EM data is beautiful and the structures are well done. The presentation however is a bit tedious to read and not described for a broad

audience, but instead targeted to a structural audience. Therefore extensive re-writing to make the paper more accessible would be warranted. Essentially this is a deep dive into a phage structure, however the in vitro ejection and the cryo-ET data make the mechanism of attachment and genome delivery a solid story. That being said, there is far too much discussion over the nitty gritty of the structural details for several of the proteins for a journal like Nature Communications.

Major

1. Structures and FSC curves etc need to be deposited in the EMDB. Therefore the FSC curves in Supplemental Figure S1 could be removed to make the paper shorter.
2. The genome release intermediates are really interesting and there is not much known on this for phages in general. Therefore, the Supplemental Figure S4 may be better off highlighted within the main text. Along with this, the authors should comment on the lack of the tail needle in these structures (likely fell off during the in vitro treatment).
3. Figure 7A and text on line 412: these are most likely to be blebbed outer membrane vesicles. Work on Vibrio phages (Hampton et al 2017 Front Microbiol, Reyes-Robles et al 2018 J Bact) and Shigella phages (Parent et al 2012 Virology, Parent et al 2014 Mol Micro) have shown this phenomenon before. It is highly unlikely these vesicles would be from lysed cells. I suggest acknowledging the previous studies and referencing them.
4. The whole discussion regarding the glycine rich loop and the putative decoration protein should be removed. While it has been shown in others that this loop is a scaffold in other phages, there is zero evidence for this in SU10 and in the absence of gp10 in the particles, this argument is very weak. Removing this would also make the text shorter and easier to read.
5. Likewise you could remove the entire description of specific residues being in different domains of the structures with better figure labels. E.g. the domains of the major capsid protein could just be labeled in the figure Supplementary S3 since the majority of the readers don't need or care about these details (same statement for the other proteins as well).
6. Over the organization seems top-down instead of focused on the biology. Why not present the capsid first, then the genome ejection intermediate, followed by the cryo-ET data showing the mechanism? That would leave a natural place for a wrap up summary. As it stands, the paper just kind of ends in an awkward way.

Minor

1. Parts of the introduction are written more for a phage/virology audience rather than a broad audience. To make this more accessible to a broad readership terms like prolate, C3 morphotype etc need some explanation.
2. The Intro reads like a list of facts without clear connection why these are important. I would suggest some transition statements and to only include the most relevant details to make this easier for the reader to digest.
3. Last line of the intro: I'd argue 70% similarity does not ensure ALL Kuraviruses do the same thing. I suggest re-wording to "most if not all".
4. Results: the specific lengths and distances of structural features do not need to be described. A broad readership would not need this information and there are scale bars in the figures.
5. Page 5 line 132. Can also point out based on your mass spec data (Figure S4C) that gp10 is indeed not associated with the phage particles.
6. Figure 7 legend: specify what time point is shown in the panel A segmentation. It is not clear how this matches the data in Supplemental Figure 7.

Reviewer #3 (Remarks to the Author):

This paper describes the structure of SU10, a podovirus phage that infects E. coli. Unlike more well-described podoviruses like P22 and T7, SU10 has a prolate head. In addition to determining the near-

atomic resolution structures of the filled and empty mature capsids, they carried out cryo-electron tomography of cells being infected with SU10, showing the phage in various stages of infection and genome injection.

A lot of data is being presented, including multiple reconstructions, tomography, fluorescence microscopy etc. Most of the structures look reasonable; however, there are missing quality indicators and some of the reported statistics and FSC curves are questionable, as described below. While many elements of SU10 are similar to other known phage structures, there are some unique features that have not been described before, including its elongated prolate head, the ordered DNA inside the head, and the structural changes in the nozzle and tail fibers upon DNA ejection. The analysis of capsid structures at different states of host binding and ejection is potentially very interesting. However, the weakness of the manuscript is that there is very little analysis beyond the description of the structures. The paper might have done better with distinct Results and Discussion sections and a more in-depth analysis of the relevance of the structures on capsid assembly, DNA packaging, receptor binding, DNA injection etc. and a better biological context for the structures.

Specific comments:

Line 54: "Core proteins" are just proteins located in the core, i.e. inside the capsid. I think the authors should specify that they are specifically referring to "pilot" or "ejection" proteins, i.e. proteins that are ejected with the genome.

Line 55: To my knowledge, phi29 does not have pilot/ejection/core proteins per se, only a terminal protein that is covalently attached to the genome. This is a rather different case and cannot really be compared to the ejection proteins described here. I suggest removing mention of phi29 here.

Lines 57-62: Some of this may be a bit too general and a better description seems necessary:

"The common components of tails ... tail or nozzle proteins": Obviously "tail proteins" are components of tails. On the other hand "Nozzle" is a rather specific term originally employed to describe structural features of the T7 tail. Maybe "A major tail protein that forms a nozzle" or something like that would be better.

"The adaptor proteins have a structure function": Redundant; in fact, ALL the tail proteins have a structural function. Maybe say something like "The adaptor proteins serve as an adaptor between the tail proteins and the connector?"

"Nozzle proteins bind to receptors": Perhaps they do, but so do receptor binding proteins variously called fibers, appendages, spikes and other terms. The main role of the nozzle seems to be in genome ejection.

Line 63: "After phage binding to a cell, the inner core proteins are ejected" See comment above. This specifically refers to ejection or pilot proteins.

Line 99: "T=4, Q=20" The preferred terms are "T_{end}" and "T_{mid}." The usage of Q came from the original definition of T numbers as $T = T_{end} = Pf^2$, where $P = h^2 + hk + k^2$, and $T_{mid} = Qf$, where Q is any integer. Logically, therefore, one would be expected to say "P=4, Q=20". However, P and Q have gone out of usage. Furthermore, T stands for "triangulation," while P and Q do not stand for anything in particular, and thus T_{end} and T_{mid} should be used instead.

There are at least three geometrically distinct hexamers in the structure: those in the end cap, those on the edge of the end cap, and those in the middle of the cylindrical part. What is the difference between the subunits that make up the different type of hexamers? Are the hexamers themselves different (e.g. curvature)? Is there a distinct angle difference between the hexamers in different environments that reflect the difference in shell curvature?

The tail needle structure is at very low (18Å) resolution. This is probably due to ambiguous alignment between the threefold needle and the sixfold tail. Did the authors try symmetry expansion followed by focused 3D classification to resolve this ambiguity? Nevertheless, the shape of the density matches a

tail needle protein AlphaFold model and is presumably correct to within its limitations.

It is very hard to judge the quality of the tomography data from the data presented in Figure S11. Maybe closeup slabs through individual particles at each time point, and/or isosurface representations would be helpful. See e.g. Hu et al 2013 Science 339, 576 for ways to represent this data better. Was subtomogram averaging attempted? This could greatly enhance the quality of the reconstruction data.

In multiple places in the methods section, the term "Apix" is used. This should be written as "Å/pixel." It probably suffices to give the pixel size to 0.01Å precision.

There are no statistics to judge the quality of the models. The authors should present FSC curves between the models and the respective maps and report the resolution at FSC=0.5. There is a table (Table S2) with model quality indicators (MolProbity score, Ramachandran etc.) but it has not been populated.

Please also provide details of appropriate portions of the map density with the models fitted in (as a supplementary figure) to indicate the quality of the map.

In the FSC curves for the virion (C5 symmetry and asymmetric) in Fig S1, the randomized curve extends higher than the unmasked curve. This is an indicator of strong bias caused by the masking. For the D10 reconstruction, the "corrected" FSC curve is identical to the unmasked curve. This is also aberrant behavior that might arise from too tight masks. These FSC curves need to be corrected (presumably by re-refining the data with a looser mask).

In Fig 1, gp20 is in gray, presumably to indicate an unknown location inside the capsid. This could be mentioned in the caption. Also, if gp20 is included in the schematic, then maybe the other "core" (ejection) proteins gp21-gp24 should also be represented.

Fig S11: See comment above regarding the tomography data.

Reviewer's comments are in blue italics, our responses in black bold font.

Reviewer #1 (Remarks to the Author):

In the MS "Tail proteins of Podoviridae phage SU10 reorganize into the nozzle for genome delivery" Siborova et al describe the structure of a podophage with a very long (very prolate) capsid and short tail called SU10. The structure of the virus particle has been investigated in its pre-infection, head-packaged state and post-infection, head-empty (nearly empty) state. A tomography of phages in the midst of infection and fluorescence imaging of the infection process has been also presented. Various parts of the virus particle have been reconstructed using cryo-EM and appropriate symmetries.

Overall, this is a big (very big) body of solid work.

However, I am very concerned about the presentation of the actual results.

Consider Fig. 1. The figure legend of panels A and B (it should be A, B here and throughout, not AB), reads

"Surface representation of cryo-EM reconstruction of virion of phage SU10 colored according to protein type..."

What I see is a _composite_ cryo-EM map, which consists of 6-7 cryo-EM maps, each obtained _separately_ using different symmetries. Somehow, these reconstructions are then put together to give a new "map", which is further segmented and rendered at some arbitrary contour levels. Neither the process of combining these maps together, nor the selection of the contour level for display of such complex objects, nor the process of segmentation and the removal "unwanted" parts is explained anywhere in the MS.

A: Thank you, we have now altered the description of Fig. 1. to state that we are showing the composite cryo-EM map, according to the reviewer's suggestion (lines 578-579):

"Surface representation of composite cryo-EM map of virion of phage SU10 colored according to protein type."

Furthermore, we have now included a new section to Materials and methods describing the preparation of the composite maps (lines 523-543):

"The composite map of the SU10 virion was prepared by combining segments from subparticle reconstructions of SU10 virion regions. Sub-particle reconstructions of regions of the SU10 virion were fitted into asymmetric reconstruction of the complete SU10 virion using Chimera (the "Fit in Map" command). The composite map was built from the following reconstructions: C5 symmetrized capsid reconstruction; portal from C12 symmetrized neck reconstruction; adaptor, long tail fibers, nozzle and short tail fibers from C6 symmetrized tail reconstruction; and C3 symmetrized tail needle reconstruction. Segments of sub-particle reconstructions were combined into the composite map using the following procedure: (1) Masks covering PDB structures of the component proteins were generated using pdb2mrc. (2) The masks were extended by six voxels and an additional four voxels of soft edge (Relion3 - relion_mask_create). (3) The sub-particle reconstructions were multiplied by the corresponding masks (Relion3 - relion_image_handler). (4) For each segment threshold the structure of the entire component with maximum resolved details was selected for display. For a short tail fiber map with decreasing resolution along the fiber, three segments with different thresholds were selected. (5) The maps of segments were normalized using CCP4 "mapmask". (6) The normalized segments were combined into the final composite map using the Chimera command "vop add onGrid". The composite map of the SU10 genome release intermediate was prepared using the same procedure from capsid reconstructed with C5 symmetry and tail reconstructed with C6 symmetry."

Now consider Fig.2. “Conformational changes of SU10 tail associated with genome release. “(AC) Cartoon representations of tail of virion (BD) and genome release intermediate of phage SU10”. Again (B, D) and (A, C). What the figure fails to mention is that a large part of the shown material are AlphaFold models. The cryo-EM map(s) was (were) too low resolution for model building.

A: We have now extended the Fig. 2 legend to clearly state how the individual parts of the structures were determined (lines 603-606):

"Tail needle and central and receptor-binding domain of the short tail fiber were modeled using AlphaFold2 multimer and fitted into the cryo-EM map. The remaining structures were built into cryo-EM reconstructions."

Furthermore, we have now extended the Materials and methods section to include a description of the structure determination or modeling of all proteins presented in the manuscript (lines 506-515):

"PDB structures were built using the program COOT⁶⁶ (portal protein, adaptor protein, residues 1-90 of long tail fiber, and proximal domain of short tail fiber), built automatically using Deep Tracer⁶⁷ (nozzle protein) or modeled using AlphaFold2⁵¹ (central and receptor-binding domains of short tail fiber and tail needle). The structures were iteratively refined in real space using the program PHENIX real_space_refine.py⁶⁸ and corrected manually in COOT and ISOLDE⁶⁹. The quality of the structures and their fit to the cryo-EM density maps was assessed using comprehensive validation in the program PHENIX. Model to map FSC curves of individual structures and examples of the fit of PDB models to cryo-EM densities are presented in Fig. S13."

We have now included a new Materials and methods section describing the modeling of phage proteins using AlphaFold2 multimer (lines 516-522):

"Structures of SU10 proteins that were not resolved in cryo-EM reconstructions to sufficient resolution to enable structural determination were predicted using a local installation of AlphaFold multimer 2.1.2. The software was installed from <https://github.com/deepmind/alphafold>, using a Dockerised environment. For prediction, databases recommended by AlphaFold authors at the time of installation (winter 2021), were used (BFD, MGnify, PDB70, PDB, PDB seqres, Uniclust30, UniProt, UniRef90)."

Furthermore, consider the attached Validation Reports of the atomic models that were built and refined against cryo-EM maps (N.B. the “full portal” report is a damaged file). First of all, the reports have “This wwPDB validation report is NOT for manuscript review” across EVERY page. Secondly, the amount of bad residues - bad fit to the map or bad geometry - is pretty staggering. In some cases, 80% (!) of the atomic model is bad (e.g. in the Nozzle C6 report). Can these atomic structures be trusted at all?

A: Due to the complexity of our PDB structures, automated validation at PDB was not possible, and manual curation by PDB staff was and is a prolonged process (1-2 months). We have now provided 11 out of 14 final validation reports (we are still waiting on the remaining 3).

The high amount of residues with a bad fit to the map was caused by PDB using too low a map threshold value to calculate the fit of a model to the map. We have now included the map to model fit statistics, recalculated using appropriate threshold values, to Tables S2 and S3. We have now included quality statistics characterising model geometry in Tables S2 and S3 - analysis of the model by MolProbity indicated that they have good quality relative to the resolution at which they were determined. Additionally, we show model to map FSC curves and examples of PDB structures in cryo-EM densities in Fig. S13.

Finally, when referring to AlphaFold models, the authors never explain or show the predicted accuracy or quality of AlphaFold models. In most figures, we do not know whether we are looking at an AF model or real structure. Fig. S9 is probably the only figure where the legend states that most of

the ribbon diagram shown is an AF model. In a way, it would be interesting to see the AF model for the whole protein.

A: We used AlphaFold2 to predict structures of tail needle and two domains of short tail fibre. We have now modified figure legends to state which parts of the structures were determined using AlphaFold2 (lines 603-6054):

"Tail needle and central and receptor-binding domain of the short tail fiber were modeled using AlphaFold2 multimer and fitted into the cryo-EM map."

Furthermore, we have now included structures coloured according to pLDDT scores (AlphaFold2 reliability score) for all structures that we modeled using AlphaFold2 (Fig. S7 and S8).

I am not sure which of the two AlphaFold multimer versions the authors used in their work, but this version:

ColabFold: AlphaFold2 using

MMseqs2 <https://colab.research.google.com/github/sokrypton/ColabFold/blob/main/AlphaFold2.ipynb> despite the authors' claim of capable of predicting multimeric structures, failed miserably on all sequences I tried (produced absolute gibberish).

However, the official AlphaFold Colab Pro

version: <https://colab.research.google.com/github/deepmind/alphafold/blob/main/notebooks/AlphaFold.ipynb> works well for moderately sized multimers. A locally installed version works even better. If reported unaltered (without refinement against the cryo-EM map), AF models must be subject to proper validation. A good correlation coefficient with the cryo-EM map tells us nothing about whether the model is real or not. The model can have clashes galore and fit the map perfectly.

A: Used AlphaFold2 multimer v2.1.2 installed locally. This information has now been included in the Materials and Methods section (lines 516-522):

"Structures of SU10 proteins that were not resolved in cryo-EM reconstructions to sufficient resolution to enable structural determination were predicted using a local installation of AlphaFold multimer 2.1.2. The software was installed from <https://github.com/deepmind/alphafold>, using a Dockerised environment. For prediction, databases recommended by AlphaFold authors at the time of installation (winter 2021), were used (BFD, MGnify, PDB70, PDB, PDB seqres, Uniclust30, UniProt, UniRef90)."

The models generated using AlphaFold2 were refined against corresponding cryo-EM maps. The description of the procedure is in Materials and Methods (lines 507-512):

"PDB structures were built using the program COOT (portal protein, adaptor protein, residues 1-90 of long tail fiber, and proximal domain of short tail fiber), built automatically using Deep Tracer (nozzle protein) or modeled using AlphaFold⁵¹ (central and receptor-binding domains of short tail fiber and tail needle). The structures were iteratively refined in real space using the program PHENIX `real_space_refine.py` and corrected manually in COOT and ISOLDE."

To provide evidence of the quality of our structures, we have now included model map correlation coefficients in the manuscript text, model to map FSCs in Fig. S13, and clashscores and other structure quality indicators in Tables S2 and S3. Furthermore, we now provide figures of the AF2 modelled structures colored according to the pLDDT scores, which indicate the local confidence of the predicted structure.

Minor concern

The Cryo-ET and fluorescence microscopy section. I am not sure what are we learning in this section. Is this process different to that in other phages? Was there a hypothesis that needed to be tested? It is difficult to write this because, obviously, a lot of work went into producing these images, but I do not know if all this work add any new information to what we already know about phage infection.

A: We consider the cryo-ET and fluorescence microscopy experiments to be essential for placing our single-particle cryo-EM structures into biological context. The tomography results provide

evidence that the formation of the tail nozzle is required for SU10 genome delivery *in vivo*. Verifying this result using the natural infection of cells was important, because the nozzle formation is a novel aspect of kuravirus infection, exceptional among podoviruses. The fluorescence microscopy studies enabled straightforward characterisation of the dynamics of SU10 infection. Furthermore, these results were appreciated by reviewers #2 and #3.

There are several other comment/suggested edits aimed at making this MS better.

The abstract needs a major overhaul.

The structure of the phage particle must be described first. Briefly. The classification and phage therapy stuff can be removed, in principle.

A: Thank you, we have now rewritten the abstract according to the reviewer's suggestion (lines 12-27):

" Escherichia coli phage SU10 belongs to the genus Kuravirus from the family Podoviridae of phages with short tails. However, in contrast to other podophages, the tails of Kuraviruses were shown to elongate upon cell attachment. We used cryo-electron microscopy of SU10 particles before and after genome ejection and cryo-electron tomography of infected E. coli cells to describe the structural changes of the phage tail that are required for its genome ejection and delivery. The virion of SU10 has a prolate head, containing genome and ejection proteins, and a tail, which is formed of portal, adaptor, nozzle, and tail needle proteins and decorated with long and short fibers. The binding of the long tail fibers to the receptors in the outer bacterial membrane is followed by the straightening of nozzle proteins and rotation of short tail fibers. In the new arrangement, the nozzle proteins and short tail fibers alternate to form a nozzle that extends the tail by 28 nm. To open the tail channel, the tail needle detaches from the nozzle proteins. Five types of ejection proteins, one of which has the predicted peptidoglycan-degradation activity, are released from the SU10 head before or together with the genome. The nozzle with the putative extension formed by the ejection proteins enables the delivery of the SU10 genome into the bacterial cytoplasm. It is likely that this mechanism of genome delivery, involving the formation of the tail nozzle, is employed by all Kuraviruses."

L. 20-21

The binding of the long tail fibers to the receptors in the outer bacterial membrane is followed by the straightening of nozzle proteins and rotation of short tail fibers by 135°.

We do not know what those things are at this point. Are they part of the phage particle or displayed on the host cell surface?

A: We have now included a sentence describing the SU10 virion structure in the abstract to clarify this statement (lines 17-21):

"The virion of SU10 has a prolate head, containing genome and ejection proteins, and a tail, which is formed of portal, adaptor, nozzle, and tail needle proteins and decorated with long and short fibers. The binding of the long tail fibers to the receptors in the outer bacterial membrane is followed by the straightening of nozzle proteins and rotation of short tail fibers."

L. 22. Prolongs -> extends

A: Accepted.

L. 23. What is a tail needle? We do not know.

A: We have now included a sentence describing the SU10 virion structure in the abstract to clarify this statement (lines 17-19):

"The virion of SU10 has a prolate head, containing genome and ejection proteins, and a tail, which is formed of portal, adaptor, nozzle, and tail needle proteins and decorated with long and short fibers."

L. 24. What is "the inner core"? A core of what?

A: We have now modified the abstract to avoid using the term "inner core". We use the term "ejection proteins" instead, and we define that they are part of the phage head.

The last sentence should be somewhere in the middle or even earlier in the abstract. After the "we show" sentence, a more detailed description of the results should be given. Then, if there is room, the study can be placed into a wider context and phage therapy can be mentioned.

A: As part of the rewriting of the abstract, we have removed the reference to phage therapy.

Results

The ring densities in the map of the empty (or partially empty) capsid shown Fig. S5 are an averaging artifact. In the raw micrograph shown in Fig. S4 we see rings all over the inside of the capsid. They are not separated into two different types. A presumably single strand of DNA can be seen.

Averaging split it into two types of rings. But neither appears to correspond to the original structure.

A: We beg to differ with the reviewer's interpretation of our results. Individual images of genome release intermediates are too noisy to determine whether they contain one or two types of ring densities. In contrast, two-dimensional reference-free class averages (example shown in Fig. 1) demonstrate that there are two types of ring densities. The 3D reconstruction of the head of the SU10 genome release intermediate contains two rings associated with each stack of capsid protein hexamers. The two rings interact with different parts of the hexamers, and therefore it is not surprising that they differ in structure.

Our manuscript contains an explanation of why the two types of rings differ in structure Fig. S9 and (lines 273-275):

"The reconstruction of the head of the SU10 genome release intermediate contains rings of putative DNA density attached to the inner face of the capsid (Fig. 1E, S9)."

and supplementary materials (lines 107-112):

"There are two types of DNA densities that interact with each H1 hexamer of capsid proteins. For the numbering of pentamers and hexamers in the SU10 capsid, see Fig 3. The first type interacts with the centers of hexamers, has a wiggly structure, and is indicated with a red arrowhead. The second type of DNA density interacts with the edges of hexamers, is straight, has a stronger density than the first type, and is indicated with a blue arrowhead."

L. 206-247. I am not sure what new about portal proteins we are learning here from this very detailed description. Please shorten and concentrate on novel aspects.

A: We have now shortened the description of the SU10 portal. The new text is focused on novel findings (lines 138-161):

" The portal complex of SU10 is embedded in the capsid at one of its fivefold vertices, where it replaces a pentamer of capsid proteins (Fig. 1AB, 4A). The portal protein of SU10 can be divided into the wing, stem, clip, crown, and barrel domains (Fig. 4B) 29. The wing domain is the largest, and is formed by eight helices and a β -sandwich containing two β -sheets of five and three antiparallel β -strands (Fig. 4B). The positively charged side-chains of Lys3 and Lys5 from the wing domain interact with the DNA that wraps around the portal complex (Fig. 4A). The two lysines are repeated twelve times in the portal complex, and thus form a high-avidity interface that is likely to bind DNA soon after the genome packaging is initiated. Anchoring the DNA end to the portal would influence the packaging of the genome into the head and its final structure. Furthermore, the wing domain mediates the interactions of the portal complex with the capsid (Fig. 4AC). Because of the mismatch of the fivefold symmetry of the capsid and twelvefold symmetry of the portal, we characterized their interactions using an asymmetric reconstruction of the neck region of the SU10 virion, which achieved a resolution of 4.6 Å (Table S2). Residues 8-44 from the wing

domain form a curved α -helix that lines the inner face of the capsid (Fig. 4AB). Additional interactions between the portal and capsid are mediated by the wing domain's stem loop (Fig. 4AB). The tunnel loop from the wing domain narrows the portal channel of SU10 to 33 Å (Fig. 4, S5). In phages T7 and P23-45, it was shown that the tunnel loops open the portal channel and thus regulate genome release^{19,30}. However, the tunnel loops in the SU10 virion and genome release intermediate have the same structure, therefore it is likely that a different mechanism ensures the genome retention of SU10 (Fig. S5).

The stem domain connects the wing and clip domains and spans across the capsid shell (Fig. 4B). The clip domain of the portal complex reaches out from the capsid and enables the attachment of the adaptor proteins (Fig. 4A)."

L. 289-291 and Fig. S8. Comparison of structures of receptor-binding domains of long tail fibers of tailed phages. Well, we are looking at chaperones (!) of LTF in tailed phages. These chaperones fall off the tip of the fiber. The entry for the cited PDB 3GUD is called "Crystal structure of a novel intramolecular chaperon" (!).

Lots of things published about those intramolecular chaperones. The cleavage site is well defined and is very conserved. See, for example

<https://www.ncbi.nlm.nih.gov/pmc/articles/PMC2666599/>

<https://pubmed.ncbi.nlm.nih.gov/17158460/>

<https://pubmed.ncbi.nlm.nih.gov/19450535/>

Furthermore, the AF model shown in Fig. S8 panel A is clearly incorrect. The three beta-sheets at the bottom of the protein MUST form a beta-helix, similar to the structure shown in panel C (and other structures described in the references above and many others!).

So, the "thing" that protrudes upwards is a chaperone that is not present in the mature fiber. It does not interact with cell surface receptors.

A: Thank you, we have now removed Fig. S8 and the incorrect speculation about LTF function from the manuscript.

L. 295-296 ...by a trimer of tail needle proteins. I think in this context the needle protein is a singular entity that contains 3 polypeptide chains. Not three tail need proteins.

A: The sentence has been modified according to the reviewer's suggestions (lines 75-77):

"The tail needle, which is formed by three polypeptide chains (gp18), protrudes from the central channel formed by the nozzle proteins (Fig. 1A-C, 2AC)."

L. 292-338. How does the structure of this tail compares to other tails? Is a similar beta-propeller found elsewhere? What does sequence conservation mapped on the structure look like?

A: We have now included a comparison to the structure of the T7 phage, however there is no sequence similarity between nozzle proteins of T7 and SU10 (lines 194-197):

"According to the convention established for phage T7, the nozzle protein of SU10 can be divided into the platform, beta-propeller, and nozzle domains. However, the nozzle proteins of SU10 and T7 exhibit no detectable sequence similarity."

Fig. S6. Panel A. Why is only a part of the nozzle model shown?

A: The aim of Fig. S5 (new numbering) is to show the dimensions of the SU10 tail channel. Indeed, parts of the nozzle proteins were removed because they extend to the front and back of the channel and they may appear to block to channel. It is stated in the figure legend that the parts of the nozzle proteins extending to the front and back of the tail in SU10 virion were removed from the display to show the channel dimensions (supplementary material):

"Parts of nozzle proteins extending to the front and back of the tail in the SU10 virion were removed from the display to show the channel dimensions."

L. 357- 359. The HxH motif is common in tail fibers and central spike proteins. See, e.g. <https://journals.plos.org/plosone/article/comments?id=10.1371/journal.pone.0211432>
<https://pubmed.ncbi.nlm.nih.gov/26295253/>
[https://www.cell.com/fulltext/S0969-2126\(11\)00469-2](https://www.cell.com/fulltext/S0969-2126(11)00469-2)
<https://pubmed.ncbi.nlm.nih.gov/22922659/>

The HxH was reported to bind Zn in T4 gp12. This is likely an artifact of the gp12 purification procedure where the protein is partially denatured in the presence of EDTA first and then refolded in the presence of Zn ions.

So, it appears that all data shows that HxH motifs in trimeric fibers and tailspikes bind a Fe ion (Fe²⁺ to be more specific).

A: Thank you, we have now included information about the nature of the ions together with the additional references (lines 241-243):

"The SU10 short tail fiber protein contains two pairs of histidines (His194 and His196, His230 and His232), which may bind Fe²⁺ ions and thus stabilise the fiber, as occurs in T4 and JUB59."

Fig. S9 is the only figure in which the figure legend states what is shown in the figure. Here in Panel A, and everywhere where a density is shown, the figure legend MUST state the contour level of the cryoEM map in standard deviations above the mean.

A: We have now included information about map contour levels to all relevant figure legends.

Fig. S10. The contour level is not stated (see Fig. S9 notes). Panel B is not an additional evidence that supports panel A. The model shown in panel A _already_ contains the predicted secondary structure information (it has been built using that information). A much more informative panel would be one that shows the reliability of AF model - the IDDT values either mapped onto the structure or as a plot, and the Predicted Aligned Error plot. This is applicable to all AF models.

A: We have now included pLDDT scores in Fig. S7 and S8 (new figure numbering) and modified Fig. S7 according to the reviewer's suggestion.

All ribbon diagram drawings must have residue numbers in strategic locations. E.g. we have no idea how to relate what's shown in panels A and B in Fig. S10.

A: We have now included indicators of selected residues in Fig. S7 and S8 (new figure numbering). In the main text figures, we have now included 1D diagrams of protein domain organisation.

Reviewer #2 (Remarks to the Author):

In the manuscript "Tail proteins of Podoviridae phage SU10 reorganize into the nozzle for genome delivery" by Šiborova et al, the authors use cryo-electron microscopy (both single particle analysis and tomography) to determine the structure of the phage. The cryo-EM data is beautiful and the structures are well done. The presentation however is a bit tedious to read and not described for a broad audience, but instead targeted to a structural audience. Therefore extensive re-writing to make the paper more accessible would be warranted. Essentially this is a deep dive into a phage structure, however the in vitro ejection and the cryo-ET data make the mechanism of attachment and genome delivery a solid story. That being said, there is far too much discussion over the nitty gritty of the structural details for several of the proteins for a journal like Nature Communications.

A: We have shortened and rewritten the manuscript to make it more accessible and attractive for a general audience.

Major

1. Structures and FSC curves etc need to be deposited in the EMDB. Therefore the FSC curves in Supplemental Figure S1 could be removed to make the paper shorter.

A: We have now removed Fig. S1.

2. The genome release intermediates are really interesting and there is not much known on this for phages in general. Therefore, the Supplemental Figure S4 may be better off highlighted within the main text. Along with this, the authors should comment on the lack of the tail needle in these structures (likely fell off during the in vitro treatment).

**A: We have now moved the reference-free 2D class averages of the SU10 virion and genome release intermediate from Fig. S4 to Fig. 1. The tail needle needs to detach from the tail so that the DNA can be released through the tail channel. This is discussed in lines 346-347:
"Subsequently, the tail needle dissociates from the tail to enable the opening of the tail channel."**

3. Figure 7A and text on line 412: these are most likely to be blebbed outer membrane vesicles. Work on Vibrio phages (Hampton et al 2017 Front Microbiol, Reyes-Robles et al 2018 J Bact) and Shigella phages (Parent et al 2012 Virology, Parent et al 2014 Mol Micro) have shown this phenomenon before. It is highly unlikely these vesicles would be from lysed cells. I suggest acknowledging the previous studies and referencing them.

**A: Thank you, we have now corrected the interpretation of the origin of vesicles and included the suggested references in the main text (lines 315-317):
"In the late stages of SU10 infection, the sample contained outer membrane vesicles with bound phage particles. The membrane vesicles can be shed by bacteria growing in planktonic culture."**

4. The whole discussion regarding the glycine rich loop and the putative decoration protein should be removed. While it has been shown in others that this loop is a scaffold in other phages, there is zero evidence for this in SU10 and in the absence of gp10 in the particles, this argument is very weak. Removing this would also make the text shorter and easier to read.

A: According to the reviewer's suggestions, we have now removed the discussion of glycine-rich loops (and Fig. S3) and shortened the discussion of the putative function of gp10 (discussion of the absence of gp10 from SU10 was requested by reviewer #2 - see below) (lines 106-111):

"The genome of SU10 encodes a protein (gp10) with a predicted immunoglobulin-like fold similar to that of the minor capsid protein of phage Epsilon15 and the N-terminal domain of head fiber protein of bacteriophage ϕ 29. However, the cryo-EM reconstruction of the SU10 head does not contain density corresponding to minor capsid proteins. It is possible that gp10 of SU10 has a different function or binds to the capsid with low affinity and was lost during phage purification."

5. Likewise you could remove the entire description of specific residues being in different domains of the structures with better figure labels. E.g. the domains of the major capsid protein could just be labeled in the figure Supplementary S3 since the majority of the readers don't need or care about these details (same statement for the other proteins as well).

A: Thank you, we have now removed the description of specific residues from the main text and included 1D diagrams of protein domain composition in Fig. 3, 4, 5, 6.

6. Over the organization seems top-down instead of focused on the biology. Why not present the capsid first, then the genome ejection intermediate, followed by the cryo-ET data showing the mechanism? That would leave a natural place for a wrap up summary. As it stands, the paper just kind of ends in an awkward way.

A: Thank you, we have now reorganised and rewritten the manuscript as requested.

Minor

1. Parts of the introduction are written more for a phage/virology audience rather than a broad audience. To make this more accessible to a broad readership terms like prolate, C3 morphotype etc need some explanation.

A: We have now shortened and modified the introduction to make it accessible to a broader audience. We included explanations of some of the terms and removed the remaining ones.

2. The Intro reads like a list of facts without clear connection why these are important. I would suggest some transition statements and to only include the most relevant details to make this easier for the reader to digest.

A: As indicated above, we have rewritten the introduction according to the reviewer's requirements.

3. Last line of the intro: I'd argue 70% similarity does not ensure ALL Kuraviruses do the same thing. I suggest re-wording to "most if not all".

A: Thank you, we have now modified the sentence according to the reviewer's suggestion (lines 60-61):

"Therefore, it is likely that this mechanism of genome delivery, involving the formation of the tail nozzle, is employed by most, if not all Kuraviruses."

4. Results: the specific lengths and distances of structural features do not need to be described. A broad readership would not need this information and there are scale bars in the figures.

A: We have now removed unnecessary numerical information from multiple places in the manuscript as requested by the reviewer.

5. Page 5 line 132. Can also point out based on your mass spec data (Figure S4C) that gp10 is indeed not associated with the phage particles.

A: Thank you, we have now included a reference to the MS data in the discussion of gp10 (lines 106-111):

"The genome of SU10 encodes a protein (gp10) with a predicted immunoglobulin-like fold similar to that of the minor capsid protein of phage Epsilon15 and the N-terminal domain of head fiber protein of bacteriophage ϕ 29. However, the cryo-EM reconstruction of the SU10 head does not contain density corresponding to minor capsid proteins. It is possible that gp10 of SU10 has a different function or binds to the capsid with low affinity and was lost during phage purification."

6. Figure 7 legend: specify what time point is shown in the panel A segmentation. It is not clear how this matches the data in Supplemental Figure 11.

A: The tomogram in Fig. 7A shows a segment of a cell 45 minutes post-infection. This information has now been included in the figure legend (lines 699-700):

"Segment of infected *E. coli* cell 45 minutes post-infection contains SU10 phages at various stages of infection cycle."

The 45 minutes time point is the same as that in Fig. S10GHI (new figure numbering).

Reviewer #3 (Remarks to the Author):

This paper describes the structure of SU10, a podovirus phage that infects E. coli. Unlike more well-described podoviruses like P22 and T7, SU10 has a prolate head. In addition to determining the near-atomic resolution structures of the filled and empty mature capsids, they carried out cryo-electron tomography of cells being infected with SU10, showing the phage in various stages of infection and genome injection.

A lot of data is being presented, including multiple reconstructions, tomography, fluorescence microscopy etc. Most of the structures look reasonable; however, there are missing quality indicators and some of the reported statistics and FSC curves are questionable, as described below. While many elements of SU10 are similar to other known phage structures, there are some unique features that have not been described before, including its elongated prolate head, the ordered DNA inside the head, and the structural changes in the nozzle and tail fibers upon DNA ejection. The analysis of capsid structures at different states of host binding and ejection is potentially very interesting. However, the weakness of the manuscript is that there is very little analysis beyond the description of the structures. The paper might have done better with distinct Results and Discussion sections and a more in-depth analysis of the relevance of the structures on capsid assembly, DNA packaging, receptor binding, DNA injection etc. and a better biological context for the structures.

A: We have shortened and rewritten the manuscript to make it more accessible and attractive for a general audience.

Specific comments:

Line 54: "Core proteins" are just proteins located in the core, i.e. inside the capsid. I think the authors should specify that they are specifically referring to "pilot" or "ejection" proteins, i.e. proteins that are ejected with the genome.

A: Thank you, we have now replaced all occurrences of "core proteins" with "ejection proteins".

Line 55: To my knowledge, phi29 does not have pilot/ejection/core proteins per se, only a terminal protein that is covalently attached to the genome. This is a rather different case and cannot really be compared to the ejection proteins described here. I suggest removing mention of phi29 here.

A: Thank you, we have now removed the reference to phi29 from this sentence (lines 43-44):
"The heads of *Podoviridae* phages contain ejection proteins, which play a role in the delivery of phage genomes into the host cells."

Lines 57-62: Some of this may be a bit too general and a better description seems necessary: "The common components of tails ... tail or nozzle proteins": Obviously "tail proteins" are components of tails. On the other hand "Nozzle" is a rather specific term originally employed to describe structural features of the T7 tail. Maybe "A major tail protein that forms a nozzle" or something like that would be better.

A: We have now rewritten the general description of podoviridae tails to avoid mention of nozzle proteins that are indeed specific only for a subgroup of these phages (lines 45-48):

"Common components of tails of podoviruses infecting Gram-negative bacteria are adaptor proteins, major tail proteins, tail needle proteins, and tail fibers. The tail fibers enable the initial binding of phages to bacteria, whereas tail needles are responsible for the penetration of the host cell's outer membrane."

"The adaptor proteins have a structure function": Redundant; in fact, ALL the tail proteins have a structural function. Maybe say something like "The adaptor proteins serve as an adaptor between the tail proteins and the connector?"

A: When modifying the introduction according to reviewer #2's request, the above-mentioned sentence was removed. Removing the sentence resolves the issue raised here.

"Nozzle proteins bind to receptors": Perhaps they do, but so do receptor binding proteins variously called fibers, appendages, spikes and other terms. The main role of the nozzle seems to be in genome ejection.

A: When modifying the introduction according to reviewer #2's request, the above-mentioned sentence was removed.

Line 63: After phage binding to a cell, the inner core proteins are ejected" See comment above. This specifically refers to ejection or pilot proteins.

A: Thank you, we have now modified the text to only use the term "ejection proteins".

Line 99: "T=4, Q=20" The preferred terms are "Tend" and "Tmid." The usage of Q came from the original definition of T numbers as $T (= Tend) = Pf2$, where $P=h2+hk+k2$, and $Tmid=Qf$, where Q is any integer. Logically, therefore, one would be expected to say "P=4, Q=20". However, P and Q have gone out of usage. Furthermore, T stands for "triangulation," while P and Q do not stand for anything in particular, and thus Tend and Tmid should be used instead.

A: Thank you, we have now modified the labelling as requested by the reviewer (lines 79-81): "The prolate capsid of SU10 is organized with fivefold symmetry and is built from 715 copies of the major capsid protein organized with the triangulation parameters $T_{end} = 4$, $T_{mid} = 20$."

There are at least three geometrically distinct hexamers in the structure: those in the end cap, those on the edge of the end cap, and those in the middle of the cylindrical part. What is the difference between the subunits that make up the different type of hexamers? Are the hexamers themselves different (e.g. curvature)? Is there a distinct angle difference between the hexamers in different environments that reflect the difference in shell curvature?

A: We differentiate six types of hexamers (Fig. 3A). We have now included additional supplementary figures (Fig. S1, S2) showing hexamer curvature and the angles at which the hexamers interact as requested by reviewer #3. Fig. S1 shows top and side views of all the hexamers (and the two types of pentamers) in the SU10 capsid together with information on their curvature. Fig. S2 shows the angles of hexamer-hexamer and hexamer-pentamer interfaces, explaining structural variations required for the formation of the prolate SU10 head. We have now added a corresponding discussion to the manuscript (lines 86-93):

"However, the formation of the prolate SU10 head also requires variation in the shape of hexamers and in the angles at which they interact. The hexamers in the caps are more bent (H4, H5; 149-150°), whereas those that form the tubular part of the capsid are relatively flat (H1, H2L, H2R, H3; 154-164°) (Fig. S2). All hexamer-hexamer and hexamer-pentamer interfaces in the caps are arched (145°) (Fig. S2A), however, the hexamers that form the tubular part of the capsid are connected by two types of interfaces, arched (145°) and more planar (160°) (Fig. S2BC). The arched interfaces enable the formation of rings of hexamers, whereas the more planar ones mediate interactions between the rings (Fig S2A)."

The tail needle structure is at a very low (18Å) resolution. This is probably due to ambiguous alignment between the threefold needle and the sixfold tail. Did the authors try symmetry expansion followed by focused 3D classification to resolve this ambiguity? Nevertheless, the shape of the density matches a tail needle protein Alphafold model and is presumably correct to within its limitations.

A: Thank you for the suggestion, we tried to determine the high-resolution structure of the tail needle using symmetry expansion, but were unsuccessful. The description of our attempt that led to the best tail needle structure is described in the Material and methods section (lines 455-462):

"Particle orientations from the C6 reconstruction of the phage tail were used to extract sub-particle images centered on the tail needle of the SU10 virion. The images of sub-particles were subjected to several rounds of 2D classifications, which enabled the selection of a homogeneous dataset. After initial auto-refinement with "relax symmetry" set to C3, 3D classification was performed without the alignment step. Particles belonging to the best class were selected for further Relion auto-refinement with imposed C3 symmetry and maximum allowed deviations from previous orientations of 10° (Fig. S11)."

It is very hard to judge the quality of the tomography data from the data presented in Figure S11. Maybe closeup slabs through individual particles at each time point, and/or isosurface representations would be helpful. See e.g. Hu et al 2013 Science 339, 576 for ways to represent this data better. Was subtomogram averaging attempted? This could greatly enhance the quality of the reconstruction data.

A: To enable better judgement of tomogram quality, we have now included high-magnification details of individual phage particles in Fig. S10 (new figure numbering). We attempted to perform subtomogram averaging of phage particles attached to the cells, however, the resulting structure was of low quality that did not enable us to obtain any new biological insights.

In multiple places in the methods section, the term "Apix" is used. This should be written as "Å/pixel." It probably suffices to give the pixel size to 0.01Å precision.

A: Thank you, both aspects have been modified according to the reviewer's suggestions.

There are no statistics to judge the quality of the models. The authors should present FSC curves between the models and the respective maps and report the resolution at FSC=0.5. There is a table (Table S2) with model quality indicators (MolProbity score, Ramachandran etc.) but it has not been populated.

A: We have now included the requested model versus map FSC curves (Fig. S13). We have now completed Tables S2 and S3 with structure quality indicators.

Please also provide details of appropriate portions of the map density with the models fitted in (as a supplementary figure) to indicate the quality of the map.

A: We have now included example portions of the map densities with fitted models (Fig. S13).

In the FSC curves for the virion (C5 symmetry and asymmetric) in Fig S1, the randomized curve extends higher than the unmasked curve. This is an indicator of strong bias caused by the masking.

A: The higher values of the randomised curve in the low resolution region are not necessarily an indication of overmasking. (We have recalculated reconstructions with expanded masks, but this feature persists.) The reconstructions achieved resolutions of 4.1 and 5.6 Å, respectively. The relatively low resolution of the reconstructions make the regions where the randomised curves are higher appear more prominent than in high-resolution reconstructions. As requested by reviewer #2, we have now removed Fig. S1 with FSC curves. Nevertheless, the FSC curves are available in the PDB validation reports.

For the D10 reconstruction, the "corrected" FSC curve is identical to the unmasked curve. This is also aberrant behavior that might arise from too tight masks. These FSC curves need to be corrected (presumably by re-refining the data with a looser mask).

A: We have now re-refined the reconstruction with a mask extended by 8 voxels and a soft edge of 5 voxels, however, the curves remain nearly identical. The similarity of the curves may be due to the use of both outer and inner masks.

In Fig 1, gp20 is in gray, presumably to indicate an unknown location inside the capsid. This could be mentioned in the caption. Also, if gp20 is included in the schematic, then maybe the other “core” (ejection) proteins gp21-gp24 should also be represented.

A: We have now removed the ejection proteins from the image in Fig. 1F.

Fig S11: See comment above regarding the tomography data.

A: We attempted to perform subtomogram averaging of phage particles attached to the cells, however, the resulting structure was of low quality and did not enable us to obtain any new biological insights.

REVIEWER COMMENTS

Reviewer #3 (Remarks to the Author):

Review of revised version of Siborova et al.

In this revised version of the manuscript by Siborova et al the authors have addressed the comments and criticisms from the previous review, and present an improved paper, with more thoughtful analysis of the data and better validation of the structures. The analysis of capsomer angles in Figs. S1 and S2 is nicely presented and the inclusion of model-to-map FSCs and details of density are welcome additions. Model statistics are included and look good. The figures are of good quality and provide a good illustration of the data.

The key finding in this work is the conformational changes that occur in the tail upon genome injection into the host. The inclusion of the tomography and fluorescence data paper provides added insights into this process.

Although the presentation of the data better than the original version, this is still a very structure-heavy paper and there is minimal discussion of the functional implications and broader context of their work. The introduction is very brief and the discussion ends rather abruptly. There is limited comparison to other phages for which the infection process has been studied in some detail, such as T7 or P22. In spite of these shortcomings, however, this work will no doubt be of great interest to anybody with an interest in bacteriophage structure.

A few concerns still remain:

There has been reclassification of viral families by the ICTV. While Podoviridae (or "podoviruses") is useful as a morphological descriptor, the correct family is probably Autographiviridae. Alternatively, these could be described as the T7-like phages. At any rate, the relationship to other podovirus phages should be clarified, since the terminology "podovirus" or "Podoviridae" may be too general.

Line 43: Presumably not all Podoviridae encode ejection proteins. At least phi29 and its relatives do not. Maybe this is referring to the T7-like phages (Autographiviridae—see note on nomenclature and taxonomy above)? However, at least some P22-like phages also encode ejection proteins. Maybe just add a modifier like "frequently" or "typically".

Lines 49-53: This description specifically pertains to T7. It is not known how general it is, or whether the DNA needs to be pulled into the host in other phages, including SU10.

Line 61: Not considering the abstract, the term "nozzle" is suddenly introduced in the last sentence of the introduction. It has not yet been described what "this mechanism" refers to.

It's worth clarifying in the legend of Fig 7 that the ribosomes and virus particles were not "segmented" as such, but were taken from the reconstructions and from published structures and placed in the tomogram at the identified positions, as briefly stated in the methods.

As mentioned in the review of the previous version, the FSC curves for the C5 and asymmetric virion and the D10 virion center were odd, suggesting a problem with masking. Simply removing those curves does not address the problem. The PDB validation reports for those reconstructions are missing, so the FSC curves are not currently presented anywhere. Although one reviewer suggested removing the FSCs altogether, I would still prefer to have them as a supplementary figure. Perhaps the editor can make an executive decision.

That being said, the map density details with models (Fig. S13) look reasonable, and suggest that most of the maps are of good quality. However, it's hard to judge the model resolution from the presented model-to-map FSC curves (typically taken as the FSC=0.5 crossover point), because the

curves do not start at FSC=1.0. This is presumably caused by unmodeled density. Perhaps this FSC curve could be calculated in a program that masks out all density that is not part of the model. In general, the model-to-map resolution at FSC=0.5 should be comparable to the half-map FSCs at 0.143. At present, the reported model resolution is consistently worse than the map resolution. For example, the virion C1 reconstruction is reported as 5.6Å, but the model is only accurate to 8.13Å, and the neck C12 reconstruction is at 3.6Å, while the model is only accurate to 6.86Å according to the statistics. This could reflect a poorly modeled structure, but is most likely related to the unmodeled density in the model-to-map comparisons.

Reviewer #4 (Remarks to the Author):

Siberova and colleagues present the structure of bacteriophage SU10, a type of Kuravirus (family of Podoviridae). Using an integrative approach also involving light microscopy and cryo-electron microscopy, the authors manage to shed light on the structure and function of these viruses. The manuscript reads well and the reported findings should be of interest to a broader audience. The authors appear to have addressed all prior reviewer comments.

Minor comments:

Please include a scale bar in Fig 1A. The use of scale bars is quite sparse throughout the manuscript and I would recommend putting more, where it makes sense.

Chain H in PDB ID 7Z4F has a lot of outliers. What is the reason for this and maybe this could still be corrected before publication?

Mention of electron density  this is not correct and applies to X-ray structures, replace with e.g. just density, cryo-EM maps are electrostatic potential maps

Supplementary figure S5A, B. The authors show the tail channel-forming proteins, but have not individually labelled these (portal, adaptor, nozzle), please label these.

Reviewers' comments are in blue italics and our responses in black bold font.

Reviewer #3 (Remarks to the Author):

Review of revised version of Siborova et al.

In this revised version of the manuscript by Siborova et al the authors have addressed the comments and criticisms from the previous review, and present an improved paper, with more thoughtful analysis of the data and better validation of the structures. The analysis of capsomer angles in Figs. S1 and S2 is nicely presented and the inclusion of model-to-map FSCs and details of density are welcome additions. Model statistics are included and look good. The figures are of good quality and provide a good illustration of the data.

The key finding in this work is the conformational changes that occur in the tail upon genome injection into the host. The inclusion of the tomography and fluorescence data paper provides added insights into this process.

Although the presentation of the data better than the original version, this is still a very structure-heavy paper and there is minimal discussion of the functional implications and broader context of their work. The introduction is very brief and the discussion ends rather abruptly. There is limited comparison to other phages for which the infection process has been studied in some detail, such as T7 or P22. In spite of these shortcomings, however, this work will no doubt be of great interest to anybody with an interest in bacteriophage structure.

A few concerns still remain:

There has been reclassification of viral families by the ICTV. While Podoviridae (or "podoviruses") is useful as a morphological descriptor, the correct family is probably Autographiviridae. Alternatively, these could be described as the T7-like phages. At any rate, the relationship to other podovirus phages should be clarified, since the terminology "podovirus" or "Podoviridae" may be too general.

A: Thank you. According to the new ICTV classification, SU10 belongs to the class *Caudoviricetes* and the genus *Kuravirus*. So far, the genus has not been assigned to any family. We agree with reviewer #3 that it may be misleading to refer to the family *Podoviridae*. Therefore, we have now included the description of the new classification of SU10 in the manuscript (lines 32-35):

"Phages from the genus *Kuravirus* belong to the class *Caudoviricetes* of phages with short non-contractile tails ¹. Kuraviruses, including the *Escherichia coli* phage SU10, are distinguished among short-tailed phages by large genomes with 75-80,000 base pairs encoding more than 50 proteins ²⁻⁴."

Furthermore, we have replaced most instances of the use of *Podoviridae* with "phages with short non-contractile tails." This included the change of the title of the manuscript to:

"Tail proteins of phage SU10 reorganize into the nozzle for genome delivery"

Line 43: Presumably not all Podoviridae encode ejection proteins. At least phi29 and its relatives do not. Maybe this is referring to the T7-like phages (Autographiviridae—see note on nomenclature and taxonomy above)? However, at least some P22-like phages also encode ejection proteins. Maybe just add a modifier like "frequently" or "typically".

A: Thank you, we have modified the manuscript according to the reviewer's suggestion (lines 48-49): "The heads of most short-tailed phages contain ejection proteins, which enable the delivery of phage genomes into the host cells."

Lines 49-53: This description specifically pertains to T7. It is not known how general it is, or whether the DNA needs to be pulled into the host in other phages, including SU10.

A: We have now re-written the corresponding introduction section to address the reviewer's comment (lines 40-54):

"The genomes of kuraviruses are accommodated in prolate heads, characterized by fivefold symmetry and elongation in the direction of the tail axis^{2,3}. The genomes of tailed phages are packaged into preformed pro-heads by molecular motors through a channel formed by the dodecamer of portal proteins, which replaces a pentamer of capsid proteins at one of the fivefold vertices of the capsid. Phage tails are attached to the portal complexes⁹. Common tail components of phages with short tails infecting Gram-negative bacteria are adaptor proteins, major tail proteins, tail needle proteins, and tail fibers¹⁰. The tail fibers enable the initial binding of phages to bacteria¹¹⁻¹³, whereas tail needles are responsible for the penetration of the host cell's outer membrane^{14,15}. The heads of most short-tailed phages contain ejection proteins, which enable the delivery of phage genomes into the host cells. The ejection proteins form a translocation complex that elongates the tail to span the bacterial cell wall¹⁶⁻¹⁹. The pressure inside the phage head enables the ejection of 30-50% of the genome into a bacterial cell²⁰⁻²². However, after equalization of the pressures inside the phage head and bacterial cytoplasm, the remainder of the DNA has to be delivered into the bacterium by another mechanism^{21,23,24}."

Line 61: Not considering the abstract, the term "nozzle" is suddenly introduced in the last sentence of the introduction. It has not yet been described what "this mechanism" refers to.

A: Thank you, we have now re-written the last paragraph of the introduction to make it easier to understand (lines 60-64):

"After binding to the host, SU10 tail, nozzle proteins, and short fibers re-arrange to form a nozzle that extends the tail. Kuraviruses share more than 70% sequence similarity in their tail proteins. Therefore, it is likely that this mechanism of genome delivery, involving the formation of the tail nozzle, is employed by most, if not all Kuraviruses."

It's worth clarifying in the legend of Fig 7 that the ribosomes and virus particles were not "segmented" as such, but were taken from the reconstructions and from published structures and placed in the tomogram at the identified positions, as briefly stated in the methods.

A: We have now included the requested information into the figure legend (lines 720-723):

"Positions of phage particles, ribosomes (EMD-13270), and the chemoreceptor array (EMD-10160) were identified using template matching as implemented in EMclarity^{64,65}, and the corresponding high-resolution structures were positioned into the tomogram."

As mentioned in the review of the previous version, the FSC curves for the C5 and asymmetric virion and the D10 virion center were odd, suggesting a problem with masking. Simply removing those curves does not address the problem. The PDB validation reports for those reconstructions are missing, so the FSC curves are not currently presented anywhere. Although one reviewer suggested removing the FSCs altogether, I would still prefer to have them as a supplementary figure. Perhaps the editor can make an executive decision.

A: We have recalculated the FSC values using wider masks and show them in Fig. S13. The shapes of the curves now behave as expected for well-refined structures.

That being said, the map density details with models (Fig. S13) look reasonable, and suggest that most of the maps are of good quality. However, it's hard to judge the model resolution from the presented model-to-map FSC curves (typically taken as the FSC=0.5 crossover point), because the curves do not start at FSC=1.0. This is presumably caused by unmodeled density. Perhaps this FSC curve could be calculated in a program that masks out all density that is not part of the model. In general, the model-to-map resolution at FSC=0.5 should be comparable to the half-map FSCs at 0.143. At present, the reported model resolution is consistently worse than the map resolution. For example, the virion C1 reconstruction is reported as 5.6Å, but the models is only accurate to 8.13Å,

and the neck C12 reconstruction is at 3.6Å, while the model is only accurate to 6.86Å according to the statistics. This could reflect a poorly modeled structure, but is most likely related to the unmodeled density in the model-to-map comparisons.

A: As suggested by reviewer #3, we have recalculated the model to FSC curves for all the structures using symmetry expanded PDB models, which resulted in higher values of the model to map FSC. We have now included updated FSC plots in Fig. S14.

Reviewer #4 (Remarks to the Author):

Siberova and colleagues present the structure of bacteriophage SU10, a type of Kuravirus (family of Podoviridae). Using an integrative approach also involving light microscopy and cryo-electron microscopy, the authors manage to shed light on the structure and function of these viruses. The manuscript reads well and the reported findings should be of interest to a broader audience. The authors appear to have addressed all prior reviewer comments.

Minor comments:

Please include a scale bar in Fig 1A. The use of scale bars is quite sparse throughout the manuscript and I would recommend putting more, where it makes sense.

A: Thank you, we have now included scale bars in figures 1, 3, 4, 5, 6, S1, S2, S3, S6, S7, S8, S9, S10, S11, S12, and cover images for movies 1 and 2.

Chain H in PDB ID 7Z4F has a lot of outliers. What is the reason for this and maybe this could still be corrected before publication?

A: We have now re-refined the structure, which resulted in improvement of the following values: Ramachandran outliers 0.98%→0.62%, Ramachandran favored 90.2→91.8, and Clashscore 17.0 →11.5. These values are listed in Table S3.

Mention of electron density  this is not correct and applies to X-ray structures, replace with e.g. just density, cryo-EM maps are electrostatic potential maps

A: Thank you, we have now removed all mentions of “electron density.”

Supplementary figure S5A, B. The authors show the tail channel-forming proteins, but have not individually labelled these (portal, adaptor, nozzle), please label these.

A: We have now included portal, adaptor, and nozzle proteins labels.